# Basal friction of Fleming Glacier, Antarctica, Part B: evolution from 2008 to 2015

Chen Zhao[1,3], Rupert M. Gladstone[2], Roland C. Warner[3], Matt A. King[1], Thomas Zwinger[4], Mathieu Morlighem[5]

[1] School of Technology, Environments and Design, University of Tasmania, Hobart, Australia

[2] Arctic Centre, University of Lapland, Rovaniemi, Finland

[3] Antarctic Climate and Ecosystems Cooperative Research Centre, University of Tasmania, Hobart, Australia

[4] CSC-IT Center for Science Ltd., Espoo, Finland

[5] Department of Earth System Science, University of California Irvine, CA 92697-3100, USA

**Abstract**

The Wordie Ice Shelf-Fleming Glacier system in the southern Antarctic Peninsula has experienced a long-term retreat and disintegration of its ice shelf in the past 50 years. Increases in the glacier velocity and dynamic thinning have been observed over the past two decades, especially after 2008 when only a small ice shelf remained at the Fleming Glacier front. It is important to know whether the substantial further speed up and greater surface draw-down of the glacier since 2008 is a direct response to ocean forcing, or driven by feedbacks within the grounded marine-based glacier system, or both. Recent observational studies have suggested the 2008-2015 velocity change was due to the ungrounding of the Fleming Glacier front. To explore the mechanisms underlying the recent changes, we use a full-Stokes ice sheet model to simulate the basal shear stress distribution of the Fleming system in 2008 and 2015. This study is part of the first high resolution modelling campaign of this system. Comparison of inversions for basal shear stresses for 2008 and 2015 suggests the migration of the grounding line ~9 km upstream by 2015 from the 2008 ice front/grounding line positions, which virtually coincided with the 1996 grounding line position. This migration is consistent with the change in floating area deduced from the calculated height above buoyancy in 2015. The retrograde submarine bed underneath the lowest part of the Fleming Glacier may have promoted retreat of the grounding line. Grounding line retreat may also be enhanced by a feedback mechanism upstream of the grounding line by which increased basal lubrication due to increasing frictional heating enhances sliding and thinning. Improved knowledge of bed topography near the grounding line and further transient simulations with oceanic forcing are required to accurately predict the future movement of the Fleming Glacier system grounding line and better understand its ice dynamics and future contribution to sea level.

## 1 Introduction

In the past few decades, glaciers in West Antarctica and the Antarctic Peninsula (AP) have experienced rapid regional atmospheric and oceanic warming, leading to significant retreat and disintegration of ice shelves and rapid acceleration of mass discharge and dynamic thinning of their feeding glaciers (Cook et al., 2016; Gardner et al., 2018; Wouters et al., 2015). Most of the West Antarctic Ice Sheet and the glaciated margins of the AP (Fig. 1a) rest on a bed below sea level sloping down towards the ice sheet interior, and the grounding lines of outlet glaciers located on such reverse bed slopes may be vulnerable to rapid retreat depending on the bedrock and ice shelf geometry (e.g., Gudmundsson (2013); Gudmundsson et al. (2012); Schoof (2007)). Once perturbed past a critical threshold, such as grounding line

retreat over a bedrock hump into a region of retrograde slope, the grounding line may continue to retreat inward until the next stable state without any additional external forcing (e.g., Mercer (1978); Thomas and Bentley (1978); Weertman (1974)). This marine ice sheet instability has been invoked to explain the recent widespread and rapid grounding line retreat of glaciers in the Amundsen Sea sector, likely triggered by increased basal melting reducing the buttressing influence of ice shelves (Rignot et al., 2014). Rapid grounding line retreat and accelerated flow in these unstable systems leads to significant increases in ice discharge and increased contribution from these marine ice sheets to sea-level rise.

The former Wordie Ice Shelf (WIS; Fig. 1b) in the western coast of AP started its initial recession in 1960s with a substantial break-up occurring around 1989, followed by continuous steady retreat (Cook and Vaughan, 2010; Vaughan and Doake, 1996; Wendt et al., 2010; Zhao et al., 2017). The former ice shelf is fed by three tributaries as shown in Fig. 1b. The Fleming Glacier (FG; Fig. 1b), as the main tributary glacier, splits into two branches: the main branch to the north and the southern branch (hereafter "southern FG"). The floating part in front of the main FG disappeared almost entirely sometime between 1997 and 2000 (Fig. 1b), and the ice front position in Apr 2008 (dark blue line in Figs. 1b and 1c, Wendt et al. (2010)) almost coincides with the latest known grounding line position in 1996 (Rignot et al., 2011a). The main branch of the FG has thinned at a rate of $-6.25 \pm 0.20$ m yr$^{-1}$ near the front from 2008 to 2015, more than twice the thinning rate during 2002-2008 ($-2.77 \pm 0.89$ m yr$^{-1}$) (Zhao et al., 2017). This is consistent with the recent findings that the largest velocity changes across the whole Antarctic Ice Sheet over 2008-2015 occurred at FG (500 m yr$^{-1}$ increase close to the 1996 grounding line) (Walker and Gardner, 2017). Time series of surface velocities along the centerline of the FG (extending ~16 km upstream from the 1996 grounding line) (Friedl et al., 2018) indicate that two rapid acceleration phases occurred: in Jan-Apr 2008 and from Mar 2010 to early 2011, followed by a relatively stable period from 2011 to 2016. During the first acceleration phase in Jan-Apr 2008, the front of the FG retreated behind the 1996 grounding line position for the first time (Friedl et al., 2018).

As a marine-type glacier system residing on a retrograde bed with bedrock elevation as much as ~800 m below sea level (Fig. 1c), the Fleming system is hence potentially vulnerable to marine ice sheet instability (Mercer, 1978; Thomas and Bentley, 1978; Weertman, 1974). The acceleration and greater dynamic thinning of the FG over 2008-2015 suggests the possible onset of unstable rapid grounding line retreat (Walker and Gardner, 2017; Zhao et al., 2017), which has been confirmed by Friedl et al. (2018). The speedup of the FG before 2008 was originally assumed to be a continuing direct response to the collapse of the Wordie Ice Shelf (Rignot et al., 2005; Wendt et al., 2010). Recent studies have suggested that the recent further glacier speedup and grounding line retreat could be a direct response to oceanic forcing (Friedl et al., 2018; Walker and Gardner, 2017). An alternative hypothesis is that the recent changes arise from feedbacks in the dynamics of the evolving glacier, possibly involving the subglacial hydrology. The examination of changes in basal shear stress distributions between 2008 and 2015 in this modelling study provides a first step in exploring possible feedback hypotheses. We explore the potential for these hypotheses in Sect. 5.

By analyzing the detailed history of surface velocities, rates of elevation change, and ice front positions from 1994 to 2016, Friedl et al. (2018) suggested that the initial ungrounding of the FG from the 1996 grounding line position (Rignot et al., 2011a) occurred during the first acceleration phase between Jan and Apr 2008 and expanded further upstream by ~6-9 km by 2014, which explained the speedup and thinning of the FG since 2008. They speculated this was mainly the result of unpinning caused by increased basal melting due to the upwelling of warm Circumpolar Deep Water (CDW). However, this study by Friedl et al. (2018) lacked direct measurements of basal melting and did not perform relevant numerical modelling of the evolution of a sub-ice ocean cavity or coupling to a cavity ocean circulation model, so it is still uncertain whether the enhanced basal melting triggered by ocean warming is the dominant reason for the ungrounding process. A positive feedback between basal sliding and basal water pressure (through friction heating) upstream of the grounding line could be

another possible factor in the glacier acceleration and grounding line retreat (Bartholomaus et al., 2008; Iken and Bindschadler, 1986; Schoof, 2010). The possibility of such a feedback, is not ruled out by Friedl et al. (2018), and is discussed further in Sect. 4.2 and Sect. 5.

In this study, we employ the Elmer/Ice code (http://elmerice.elmerfem.org/) (Gagliardini et al., 2013), a three-dimensional (3D) full-Stokes ice sheet model, to solve the Stokes equations over the whole WIS-FG catchment. Our implementation of the model solves the ice flow equations and the steady-state heat equation (Gagliardini et al., 2013; Gladstone et al., 2014). We also infer the basal shear stress using an inverse method (e.g., Gillet-Chaulet et al. (2016); Gong et al. (2017)).

In the first part of this study (Zhao et al., companion paper), we explored the sensitivity of the inversion for basal shear stress to: enhancement of ice deformation rates, bedrock elevation data, the ice front boundary condition, and initial model assumptions about englacial temperatures. In the current paper, we adopt the three-cycle spin-up scheme of Zhao et al. (companion paper) to derive the distributions of basal shear stress in 2008 and 2015. We present the observational data in Sect. 2 and our methods in Sect. 3. We compare the resulting basal shear distributions for 2008 and 2015 and their connections with driving stress and basal friction heating in Sect. 4.1 and Sect. 4.2. The height above buoyancy for the two epochs is computed in Sect. 4.3 as an independent guide to grounding line changes. Through comparison of basal shear stress and height above buoyancy between 2008 and 2015, we analyze the stability of the grounding line in this period and discuss ongoing marine ice sheet instability and direct oceanic forcing as possible reasons for the speed-up of the FG in Sect. 5.

## 2 Observational Data

### 2.1 Surface elevation data in 2008 and 2015

The surface elevation dataset for 2008 (DEM2008; Fig. 2a) from Zhao et al. (companion paper) plays a central role here. To estimate the surface topography in 2015 (DEM2015; Fig. 2a), we generated the average surface-lowering rate during 2008-2015 for the fast flow regions (surface velocity in $2008 \geq 20$ m $yr^{-1}$) by using the hypsometric model for elevation change described in Zhao et al. (2017) for the same period. The DEM2015 was then generated from DEM2008 by applying these ice thinning rates from 2008 to 2015. For the area with velocities $< 20$ m $yr^{-1}$, we assume the DEM in 2015 remains the same as that in 2008.

### 2.2 Bed elevation data

The bed topography plays a significant role in simulation of basal sliding and ice flow distribution for fast-flowing glaciers (Zhao et al., companion paper), and also in interpreting the grounding line movement precisely (De Rydt et al., 2013; Durand et al., 2011; Rignot et al., 2014). Zhao et al. (companion paper) investigated the sensitivity of the basal shear stress distribution to three bedrock topography datasets. The bedrock dataset, bed_zc (Fig. 2b), with higher accuracy and resolution, was suggested as the most suitable for modelling the WIS-FG system. Recall that bed_zc is computed by:

$$\text{bed\_zc} = S_{2008} - H_{mc} \tag{1}$$

where $S_{2008}$ is the surface elevation in 2008 combined from two DEM products as discussed in Zhao et al. (companion paper), and $H_{mc}$ is the ice thickness data with a resolution of 450 m combined from the ice thickness data computed using a mass conservation method for the regions of faster flow (Morlighem et al., 2011; Morlighem et al., 2013), and ice thickness from Bedmap2 for other regions (Fretwell et al., 2013). A complete description is given by Zhao et al. (companion paper).

### 2.3 Surface velocity data in 2008 and 2015

We use the same velocity data for 2008 as in Part A of this study (Zhao et al., companion paper), which is from the InSAR-based Antarctic ice velocity dataset MEaSUREs (version 1.0) produced by Rignot et al. (2011c) from fall 2007 and/or 2008 measurements over the study area. The 2008 velocity dataset has a resolution of 900 m and the uncertainties over the study region range from 4 m yr$^{-1}$ to 8 m yr$^{-1}$. For 2015, we adopt the velocity data extracted from Landsat 8 imagery with a resolution of 240 m and errors ranging from 5 m yr$^{-1}$ to 20 m yr$^{-1}$ (Gardner et al., 2018). The velocity dataset for 2015 has a full coverage over the WIS-FG domain, while the velocity in 2008 has no data in the gray area in Fig. 1b.

### 2.4 Other datasets

The steady state temperature field is simulated from an initial temperature field, linearly interpolated between upper and lower ice surfaces, which leads to robust inversion results as demonstrated by Zhao et al. (companion paper). The surface temperature is constrained by yearly averaged surface temperature over 1979-2014 computed from RACMO2.3/ANT27 (van Wessem et al., 2014) and the basal temperature is initialized to pressure melting temperature. The temperature simulations utilize the spatial distribution of geothermal heat flux estimated by Fox Maule et al. (2005) and the simulated basal frictional heating.

Our DEM is an ellipsoidal WGS84 system and hence a height of 0 m does not refer to sea level. An observed sea level height of 15 m (WGS84 ellipsoidal height) in Marguerite Bay (Zhao et al., companion paper) was taken to compute the sea pressure on the ice front.

### 3 Method

The modelling method using Elmer/Ice presented in Part A of this study (Zhao et al., companion paper) is adopted here, including the mesh generation, mesh refinement, model parameter choices and boundary conditions. The simulations for both 2008 and 2015 retain the same assumptions about the ice-covered domain, namely a common spatial extent with fixed ice front location, and the assumption that all the ice is grounded. The ice front position is assumed to coincide with the 1996 grounding line position (Rignot et al., 2011a). This assumption might be incorrect for the main branch of the FG, and we evaluate it based on the deduced floating area where the inferred basal shear stress is lower than a threshold, which is discussed in Sect. 4.1. It is very clear from satellite imagery that in 2008 a small ice shelf is still present in front of the southern FG and the Prospect Glacier (hereafter PG) (Fig. 1c). In 2015 the ice shelf in front of the southern FG has disappeared, while the floating part of the PG has retreated in the east and re-advanced in the west (Fig. 1c). However, we don't include the floating parts of the southern FG and PG in either epoch in this study, owing to the lack of the ice shelf thickness data.

We follow the three-cycle spin-up scheme (Zhao et al., companion paper) and simulate the basal shear stress $\tau_b$ in 2008 and 2015 with the linear sliding law:

$$\tau_b = -Cu_b \tag{2}$$

Here $C$ is the basal friction coefficient, a variational parameter in the inversion procedure, and $u_b$ is the basal sliding velocity.

There are two key differences between the data used for the 2008 and 2015 inversions: increased surface velocity and changed ice geometry, namely a thinner glacier in 2015 compared to 2008 due to dynamic thinning. To explore their relative impacts, we carry out an additional inversion with the geometry from 2008 but the surface velocity from 2015 (see Sect. S1 in the supplementary material). We find that both geometry variations and velocity changes are important to the inverted basal stress condition.

To explore the relationship between the basal shear stress and local gravitational driving stress $\tau_d$, the gravitational driving stress is also computed for both epochs:

$$\tau_d = \rho_i g H |\vec{\nabla} z_s| \tag{3}$$

where $\rho_i$ is the ice density, $g$ is the gravitational constant, $H$ is the ice thickness, and $|\vec{\nabla} z_s|$ is the gradient of the ice surface elevation. Considering the snow and firn on the ice surface, we apply a relatively low ice density of 900 kg m$^{-3}$ following Berthier et al. (2012).

Hoffman and Price (2014) found a positive feedback between the basal melt and basal sliding through the frictional heating for an idealized mountain glacier using coupled subglacial hydrology and ice dynamics models. To explore possible effects of changes of basal frictional heating between 2008 and 2015, we compute the friction heating ($q_f$) generated at the bed:

$$q_f = \tau_b u_b \tag{4}$$

To explore the possible flow path of subglacial water beneath the FG, we calculate hydraulic potential at the bed, since its negative gradient determines subglacial flow direction. The hydraulic potential, $\Phi$, expressed in equivalent metres of water, is given by:

$$\Phi = (z_s - z_b) \frac{\rho_i}{\rho_{fw}} + z_b \tag{5}$$

where $\rho_{fw}$ is the fresh water density (1000 kg m$^{-3}$), and $z_s$ and $z_b$ are the surface and bed elevations, respectively. Here we assume that the water pressure in the subglacial hydrologic system is given by the ice overburden pressure, which is equivalent to assuming that the effective pressure at the bed, $N$, is zero (Shreve, 1972).

Height above buoyancy ($Z_*$) is an indicator of how close to floatation a marine-based glacier is, which is relevant to the glacier's evolution and additionally helps identify likely floating regions. $Z_*$ is related to the effective pressure $N$ at the bed by the relationship:

$$N = \rho_i g Z_* \tag{6}$$

In this study, we use a simpler hydrostatic balance based on sea level with the relationship:

$$Z_* = \begin{cases} H, & if\ z_b >= z_{sl} \\ H + (z_b - z_{sl}) \frac{\rho_w}{\rho_i}, & if\ z_b < z_{sl} \end{cases} \tag{7}$$

where $\rho_w$ is the density of ocean water and $z_{sl}$ is the sea level. This expression for $Z_*$ assumes a perfect connectivity of the basal hydrology system with the ocean. This is appropriate for the present study where we are exploring the degree of grounding of the fast flowing regions of the FG over the downstream basin.

## 4 Results

### 4.1 Comparison of basal shear stress and driving stress in 2008 and 2015

We obtain the spatial distributions for basal shear stress, $\tau_b$ (Figs. 3a, 3b), and basal velocity of the WIS-FG system for 2008 and 2015 using an inverse method to determine the basal friction coefficient, $C$, with the geometry and velocity data described above. Although low-resolution estimation of basal shear stress has been carried out for the whole Antarctic Ice Sheet (Fürst et al., 2015; Morlighem et al., 2013; Sergienko et al., 2014), this is the first application of inverse methods to estimate the basal friction pattern of the Fleming system at a high resolution and use the full-Stokes equations.

In 2008 the main FG shows some sticky spots of high basal shear stress close to the ice front (Fig. 3a). The backstress exerted by these sticky spots with $\tau_b > 0.01$ MPa (shown in Fig. S3) is $\sim 3.42 \times 10^{11}$ N, while immediately upstream a region of low basal stress covers most of the

downstream bedrock basin, returning to more typical values (~0.05-0.53 MPa) ~9 km from the ice front. In contrast, the basal friction at the front of the southern FG is low, with more typical values ~2 km upstream. By 2015, the high friction spots near the FG ice front have disappeared while in the downstream basin the region of low basal shear stress already seen in 2008 is more extensive and even lower in value (Fig. 3b). This is consistent with the observed speed-up from 2008 to 2015. Further upstream in this basin, and over the ridge between the downstream and upstream basins, the basal shear stress does not change much between the two epochs (Fig. 3c).

To explore the ice dynamics evolution from 2008 to 2015, we present the ratio of basal shear stress $\tau_b$ to driving stress $\tau_d$ (hereafter referred as "RBD") in Figs. 3d, 3e, which can provide insight into the dynamical regime (Morlighem et al., 2013; Sergienko et al., 2014). In particular, it provides an indication whether the driving stress is locally balanced by the basal shear or whether there is a significant role for membrane stresses and a regional momentum balance. We designate the region with $\tau_b$ < 0.01 MPa or RBD < 0.1 as a "low friction" area, potentially indicative of flotation, i.e. ungrounded ice.

The high basal shear stress spots inferred by the inversion at the front of the main branch of the FG in 2008 (Fig. 3a) may be artefacts due to uncertainties from the ice thickness, local bed topography, local sea level, ice mélange backstress, and the ice front position (as discussed in Zhao et al. (companion paper)). Sensitivity to such uncertainties was explored in Zhao et al. (companion paper), and the adjustments of ice front boundary condition with a higher sea level of 25 m or an advanced ice front position showed a decrease in the basal friction coefficients near the ice front, but did not completely remove these high basal friction spots. This implies that the front of the FG in 2008 might still be partly grounded on the 1996 grounding line due to the presence of real pinning points.

As expected, the gravitational driving stress of this system shows no significant changes from 2008 to 2015, except for the front of PG (Fig. 3f). In 2015, the boundaries of the zone in the main FG with $\tau_{b2015}$ < 0.01 MPa (blue lines in Fig. 3b) or $RBD_{2015}$ < 0.1 (red lines in Fig. 3e) have some similarity to the deduced grounding line position of the FG in 2014 from Friedl et al. (2018) (white dots in Figs. 3 and 4). The differences with that study are around the southern and eastern parts, but the blue and red boundaries fit the bedrock ridges in the present study (Figs. S2b), while the white points fit the corresponding bedrock topography data used by Friedl et al. (2018). This comparison confirms the significant role of bedrock topography in determining the grounding line position. Around the eastern part of the region within which velocities > 1500 m $yr^{-1}$ (Fig. 3b), the low basal friction area in this study extends ~1-3 km further upstream than the estimated grounding line in 2014 (Friedl et al., 2018).

Comparison of basal shear stress between 2008 and 2015 (Fig. 3c) shows a significant decrease from 2008 to 2015 in fast flowing regions (velocity > 1500 m $yr^{-1}$) at the front of the FG. A similar pattern occurred at front of the PG and the southern FG. For the northern section of the southern FG, the grounding line has retreated by ~2 km in 2008 from the last known grounding line position in 1996 (Rignot et al., 2011a) (Fig. 3a), which is reasonable considering that the northern section of the ice front has retreated ~2 km behind the 1996 grounding line position (Fig. 1c). However, it is not clear whether the southern section of the southern FG has also retreated in 2008 as indicated in Fig. 3a, and whether the floating area has expanded ~3 km further inland in 2015 based on the decreased basal shear stress from 2008 (Fig. 3a) to 2015 (Fig. 3b). Similarly, it is also hard to estimate the possible grounding line positions of the PG based from the inferred basal shear stress in both epochs. That is because we did not account for the normal stress of the remnant small ice shelf at the front of the southern FG and the PG (Fig. 1c) in the inverse modelling. The surface lowering in DEM2015 for the PG could also be an artefact since no observations were available for the PG when building the hypsometric model that generates the DEM2015 (see inset map in Fig. 2a; Zhao et al. (2017)).

## 4.2 Basal melting and subglacial hydrology

Increases in subglacial water pressure could contribute to lower basal shear stress and higher basal sliding at the base of the FG, potentially through the positive hydrology feedback mentioned earlier. That feedback mechanism can be summarized simply: a general acceleration of glacier flow (for example due to a backstress reduction from ice shelf collapse or unpinning from a sticky spot) can lead to increased basal sliding in regions where the basal shear stress almost remains unchanged (for example in the FG trunk above the downstream basin (Figs. 3a-c). This increases friction heating and basal melt water generation, which - as suggested by Hoffman and Price (2014) - may increase the effective basal water pressure downstream, thereby increasing sliding speeds (Gladstone et al., 2014; Hoffman and Price, 2014). Since the reduction of effective pressure is the key process to enhance sliding, this positive feedback is dependent on a positive feedback of melt water generation to water pressure. This dependence can break down when there is sufficient basal water to generate efficient drainage channels (Schoof, 2010). However, such efficient channelization in the subglacial hydrologic system is typically associated with seasonal surface meltwater pulses reaching the bed (Dunse et al., 2012), a process that is not expected to occur for Fleming Glacier (Rignot et al., 2005).

Basal melt water arises from two main sources in polar regions: either surface melt water draining into the subglacial hydrologic system via crevasses or moulins or in-situ melting at the bed (Banwell et al., 2016; Dunse et al., 2015; Hoffman and Price, 2014). However, the amount of surface melt water in the WIS-FG region is not thought to be sufficient to percolate to the base (Rignot et al., 2005), so we take basal melting due to the friction heat and geothermal heat flux as the only source of subglacial water. The geothermal heat flux in the fast flowing regions of our study area (Fox Maule et al., 2005) is two orders of magnitude smaller than the friction heating at the base, leaving friction heating as the dominant factor in generating basal melt water.

To explore the potential subglacial water sources and the likely flow directions, we plot the frictional heating (Figs. 4a, 4b), the contours of hydraulic potential ($\Phi$; Figs. 4d, 4e), and the basal temperature relative to the pressure melting point (Figs. 4g, 4h) for both epochs. Friction heating due to sliding at the bed (Figs. 4a, 4b) provides a basal melt water source where ice is at pressure melting point, which is the case for the fast flow regions of the FG (see the basal temperature relative to the pressure melting point in Figs. 4g, 4h), while the gradient of the hydraulic potential (Figs. 4d, 4e) indicates likely water flow paths at the ice-bed interface. The frictional heat generated at the base is high where both basal shear stress and basal sliding velocities are high. The modelled friction heating in both 2008 and 2015 (Figs. 4a, 4b) extends as far as the upstream basin under the FG, indicating high basal melt rates in this region (a heat flux of 1 W m$^{-2}$ could melt ice at the rate of 0.1 m yr$^{-1}$ in regions at the pressure melting temperature). The highest friction heating is generated over the bedrock rise between the FG upstream and downstream basins, where the most melt water will be produced and this will be routed towards the downstream basin given the gradient of hydraulic potential in this region (Figs. 4g, 4h). Hence it is a major source of basal water for the downstream basin. This could explain the low basal friction downstream, while the increase in heating between 2008 and 2015 (Fig. 4c) could further enhance the basal sliding in the fast-flowing regions, contributing to the observed accelerations. Both the hydraulic potential and frictional heating could help to understand the mechanism behind the rapid acceleration and surface draw-down of the FG, which is further discussed in Sect. 5.

## 4.3 Height above buoyancy

We compute the height above buoyancy, $Z_*$, for 2008 and 2015 for the FG based on Eq. (7) with a sea level of 15 m (Figs. 5a, 5b). To allow for the over- or under-estimation of $Z_*$ owing to uncertainties from the topography data, ice thickness, ice density and the sea level applied

above, we suggest that the areas where $Z_* < 20$ m might be floating, while including areas where $Z_* > -20$ m in Fig. 5.

In 2008 a low height above buoyancy (Fig. 5a) is only found near the 1996 grounding line position in the downstream basin, which indicates that ungrounding of the main FG may not have started or only just commenced in 2008. In 2015, the area close to flotation with $Z_* < 20$
340 m (taken as an upper limit) has expanded, reaching about 9 km upstream (magenta lines in Fig. 5b), which broadly coincides with the estimated grounding line in 2014 (Friedl et al., 2018) except for an almost encircled patch with slightly higher $Z_*$ (20-30 m). The implications of the different $Z_*$ from 2008 and 2015 are a small FG grounding line retreat from 1996 to 2008 but significant retreat from 2008 to 2015. Uncertainty in the predicted
grounding line in 2015 is significant, but a new position ~9 km upstream is likely.

In addition to the main branch of the FG, its southern branch and the PG also show an expansion of the region in which $Z_*$ is close to zero, which indicates possible grounding line retreat. However, the DEM2015 used to compute $Z_*$ has large uncertainties in the southern branch of FG and PG, since the surface lowering in DEM2015 for those regions could be
artefacts due to the lack of observations as mentioned above  (see inset map in Fig. 2a; Zhao et al. (2017)). Therefore, it is hard to determine the current grounding line locations for those two glaciers.

Changes in $Z_*$ from 2008 to 2015 suggest the creation of an ungrounded area consistent with the area of very low modelled basal shear stress shown in Figs. 3a and 3b. This change in area
close to floating, defined by $Z_* < 20$ m, constitutes additional evidence supporting the hypothesis of rapid grounding line retreat over 2008 to 2015 and the likely grounding line positions of the FG in both epochs.

## 5 Discussions

The sticky spots of high basal shear stress near the terminus of the FG in 2008 might be
artefacts, but the possibility that this high friction area is a real feature due to some pinning points is not excluded. If the high basal resistance spots are artefacts, ungrounding of this region in early 2008 is less viable as an explanation for an abrupt increase in ice flow speed, since the loss of backstress would be more gradual. In this case, positive feedbacks, such as the marine ice sheet instability or the basal melt feedback, are even more likely to explain the
FG's recent behavior. If the sticky spots are real features, the implication is that the ice front was at least partly grounded in early 2008. This interpretation is consistent with the relatively high bedrock topography near the ice front compared to upstream (Fig. 1c). Friedl et al. (2018) proposed that the grounding line of the FG after Jan-Apr 2008 must have been located upstream of the 1996 grounding line from their interpretation of abrupt surface acceleration
detected around the same period. This is also confirmed by the fact that the glacier front had retreated behind the 1996 grounding line during the acceleration phase (Friedl et al., 2018). However, it is possible that this grounding line retreat occurred after Jan 2008, when our DEM2008 was acquired. The analysis of height above buoyancy for DEM2008 and inferred basal shear stress in 2008 support the main FG being grounded close to the ice front and
hence near the 1996 grounding line location. Given the uncertainties of grounding line position in 1996 (several kilometres) (Rignot et al., 2011a) and uncertainty about interpreting the frontal high basal friction area in this study, the exact grounding line position in January 2008 is somewhat uncertain. Improved bed topography/ice thickness data and accurate historic ice front position are necessary to interpret the precise grounding line position in
2008. Detailed bathymetry of the relevant location might become available if the ice front of the FG retreats in future.

The disappearance of the inferred high basal shear region (possible physical pinning points) near the FG front between 2008 and 2015 is a possible trigger for the sudden acceleration and increased surface lowering of the FG during this period. The increased flux of ice, combined

with the changed glacier geometry, suggests the substantial grounding line retreat, which agrees with two recent studies (Friedl et al., 2018; Walker and Gardner, 2017). The timing of the acceleration, which occurred in Jan-Apr 2008 (Friedl et al., 2018), suggests that the loss of this basal resistance occurred shortly after the first epoch we analyzed (Jan 2008). Given the low basal friction already present over most of the downstream basin (a possible cavity

proposed by Friedl et al. (2018)), one would expect the loss of the localized friction near the ice front to promptly result in an increase in velocity over the entire low-friction region. This is consistent with the near uniform increase in velocity in Apr 2008 for a region 4-10 km upstream of the 1996 grounding line reported by Friedl et al. (2018).

For a glacier lying on a retrograde slope in a deep trough, the grounding line may be

vulnerable to rapid retreat without any further change in external forcing, once its geometry crosses a critical threshold, which is the marine ice sheet instability hypothesis (e.g., Mercer (1978); Thomas and Bentley (1978); Weertman (1974)). A similar theory has been proposed on the prospective rapid retreat of Jakobshavn Isbræ in West Greenland without any trigger after detaching from a pinning point (Steiger et al., 2017). The FG grounding line in early

2008 may have experienced a retreat after moving across the geometric pinning points near the front, and then retreated further to the position about 9 km upstream in the FG downstream basin by 2015. This has been proven by Friedl et al. (2018), and they also suggested that a further stage of grounding line retreat of the FG may have happened between Mar 2010 and early 2011. A similar ungrounding process has been detected in the Thwaites,

Smith and Pine Island Glaciers from 1996 to 2011 (Rignot et al., 2014).

The current grounding line of the FG (Friedl et al., 2018) appears to be on the prograde slope of the bedrock high between the FG downstream and upstream basins. With the establishment of an ocean cavity under the new ice shelf we can expect that ocean-warming driven basal melting will further modify the thickness of the recently ungrounded ice. If the system

remains out of balance and continues to thin, the grounding line could eventually move across this bed obstacle. If this occurs, the grounding line is then likely to retreat rapidly down the retrograde face of the FG upstream basin, likely to be accompanied by further glacier speed up and dynamic thinning.

Walker and Gardner (2017) attribute the significant increase in observed ice velocity and drop

in surface elevation from 2008 to 2015 to increased calving front melting caused by incursion of relatively warm Circumpolar Deep Water (CDW). The CDW flows onto the continental shelf within the Bellingshausen Sea, penetrating into Marguerite Bay, driven by changes in regional wind patterns resulting from global atmospheric circulation changes (Walker and Gardner, 2017). Friedl et al. (2018) also explain the unpinning from the 1996 grounding line

position in 2008 and further landward migration of the grounding line in 2010-2011 with the same mechanism, namely the increased basal melting due to ocean warming. This explanation appears consistent with the finding that the acceleration, retreat, and thinning of outlet glaciers in the Amundsen Sea Embayment (ASE) are triggered by the inflow of warm CDW onto its continental shelf and into sub-ice-shelf cavities (Turner et al., 2017). However, the

floating parts of the FG remained negligible in 2008 as indicated in Sect. 4.3 (Fig. 5a). The speedup and ungrounding occurring in the ASE glaciers was a direct response to significant loss of buttressing caused by ice shelf thinning and grounding-line retreat (Turner et al., 2017). When the CDW incursions started in the ASE, the floating parts of ASE glacier systems were much larger than the residual ice shelf of the Fleming system in 2008. After the

recent changes the newly floating region of the FG has an area of ~60 km$^2$, based on the estimated 2014 grounding line from Friedl et al. (2018) and the 2016 ice front position in this study, which is consistent with our height above buoyancy analysis for 2015 (Fig. 5b). So, significant buttressing reduction is not likely to have occurred on the FG during the rapid acceleration of 2008, but further changes to the FG after 2015 may resemble ASE glacier and

ice shelf systems more closely. No direct measurements are available to confirm the direct effect of the frontal or basal melting on the FG grounding zone over this period, nor have previous studies attempted to quantify the amount of melting required to drive significant FG

grounding line retreat. The ocean-driven basal melting at the ice shelf front or base may have contributed to grounding line retreat, or the reduction of the frontal high basal shear zone, but establishing this as the main cause would require further quantification of the cause-effect link.

Ongoing thinning as a result of backstress reduction following the collapse of the WIS is another possible cause for the recent ungrounding. The WIS evolved from an embayment-wide ice shelf in 1966 to smaller individual remnant ice shelves in 1997 (Fig. 1b) (Cook and Vaughan, 2010; Wendt et al., 2010). The floating part of the FG in particular was in the form of an ice tongue in 1997 (Cook and Vaughan, 2010), and as such would likely have imposed much lower backstress on the grounded part. Point measurements indicate that the FG accelerated by 40-50% between 1974 and 1996 (Doake, 1975; Rignot et al., 2005). If this acceleration was a response to loss of buttressing, the FG system may have been out of equilibrium, and losing mass, since before 1996. If the increased velocity in response to shelf collapse was maintained over time, maintaining persistent thinning, eventual ungrounding of the bedrock high where the 1996 grounding line was located would occur independently of ocean-induced increased shelf melt. The recent accelerations and enhanced thinning (Friedl et al., 2018; Gardner et al., 2018; Walker and Gardner, 2017) may indicate an ongoing response to the WIS collapse, amplified by positive feedbacks within the FG system.

Rapid sliding at the base is dependent on the presence of a sub-glacial hydrologic system. Evidence suggests that increased basal water supply could accelerate basal motion of both mountain glaciers (Bartholomaus et al., 2008) and ice sheets (Hoffman et al., 2011), presumably by changing the subglacial water pressure or bed contact, and further contribute to grounding line retreat of marine-based glaciers. Jenkins (2011) has also suggested that subglacial water emerging at the grounding line can enhance local ice shelf basal melt rates by driving buoyancy driven plumes in the ocean cavity. The rapid sliding and high friction heating in the upstream FG (Figs. 4a, 4b), together with the direction of the hydraulic potential gradient (Figs. 4d, 4e), provide evidence for an extensive active hydrologic system beneath the FG, which might already have been enhanced by the previous significant WIS collapse that occurred before 2008.

High basal friction heating in the fast flowing regions of the FG is the main source of meltwater flowing into the FG downstream basin. It is also clear that the friction heating in 2015 was greater than in 2008 in the upstream basin (Fig. 4c), with the increase in basal meltwater production peaking over the bedrock rise between the downstream and upstream basins (see Sect. S2 and Fig. S4). The plateaus in hydraulic potential in both downstream and upstream basins of the FG suggest the possibility that basal water may accumulate in those regions, or at least show a low throughput. The downstream plateau appears to be fed by a large frictional heat source over the ridge between the downstream and upstream basins in addition to flow from further inland, while the upstream plateau appears to be fed by an extensive upstream region of basal melting. There might be some pooling of water in those plateaus in 2008, but the inferred basal shear stress (Fig. 3a) and the height above buoyancy (Fig. 5a) indicate that those regions should still remain grounded. According to our hydraulic potential calculations, outflow from the upstream plateau region is likely to be predominantly in the direction of the downstream basin, but future outflow across the shallow saddle in hydraulic potential towards the southern branch of the FG cannot be ruled out, since the evolution of the potential responds to the changing elevation, as can be seen in Fig. 4f.

The further abrupt speed-up events that occurred in 2010-2011 reported by Friedl et al. (2018) could have several potential causes in addition to the previously proposed mechanism of a direct response to ocean-induced melting (Walker and Gardner, 2017). One possibility is an outburst of subglacial water from the upstream basin after building up over years to decades in response to increased sliding and friction heating and progressive lowering of the ice surface. Another possibility is local unpinning near the retreating grounding line: ungrounding from pinning points may cause a step reduction in basal resistance. This unpinning could be a feature of ongoing thinning in response to WIS collapse, as discussed

above. Another possibility could be positive feedbacks in the subglacial hydrologic system – rapid change may result from the direct feedback between changes in sliding speed, friction heat and basal water production.

The height above buoyancy is an indicator for the vulnerability of marine-based grounded ice to dynamic thinning and acceleration. The area with $Z_* < 20$ m in 2015 has shown that the downstream basin is currently ungrounding and this may continue until the grounding line finds a stable position on the prograde slope separating the two major basins. More thinning would be needed to destabilise the upstream basin, and it is hard to estimate how much forcing would be needed to push the grounding line into the upstream basin boundary. If the retrograde slope of the upstream basin is reached, further rapid and extensive grounding line retreat would be expected. A clear decrease can be seen in $Z_*$ from 2008 (red in Fig. 5a) to 2015 (dark red in Fig. 5b) in the upstream basin (around the 2008 velocity contour of 1000 m yr$^{-1}$), indicating the potential vulnerability of the FG to continued ice mass loss. The surface lowering rate between 2008 and 2015 in this region is ~4.6 m yr$^{-1}$ (Zhao et al., 2017). If this thinning rate continues, the ice in regions with $Z_*$ of 200-300 m would be expected to unground in ~45-65 years. This could take a longer or shorter period since the future thinning rate cannot be expected to remain constant.

In the absence of precise and accurate knowledge of bed topography and ice shelf/stream basal processes, the dominant cause of the recent FG ungrounding cannot be determined. Further research is necessary to better understand the dominant mechanisms.

**6 Conclusions**

We used a full-Stokes ice dynamics model (Elmer/Ice) at high spatial resolution to estimate the basal shear stress, temperature and friction heating of the Wordie Ice Shelf-Fleming Glacier system in 2008 and 2015. Both increased surface velocity and surface lowering during this period are important for the calculation of basal shear stress.

Decreased basal friction from 2008 to 2015 in the Fleming Glacier downstream basin indicates significant grounding line retreat, consistent with change in the suggested floating area based on the geometry in 2015 and the deduced grounding line in 2014 from Friedl et al. (2018). Grounding line retreat also occurred on the southern branch of the FG. Our height above buoyancy calculations also indicate the FG downstream basin was close to flotation in 2015 and is vulnerable to continued ice thinning and acceleration.

Pronounced basal melting driven by oceanic warming in Marguerite Bay may have triggered the ungrounding of the Fleming Glacier front in early 2008, as previously suggested by Walker and Gardner (2017) and Friedl et al. (2018), but ongoing thinning following the collapse of Wordie Ice Shelf may also provide an explanation. In either case, feedbacks in the subglacial hydrologic system may provide the dominant mechanism for rapid increases in basal sliding and ongoing ungrounding. The derived basal shear stress distributions suggest a major influence could have been the ungrounding of some sticky spots of higher basal shear near the ice front of the main Fleming Glacier, as basal friction under most of the region considered afloat by 2015 was already low in 2008 (a possible subglacial cavity).

The marine-based portion of the Fleming Glacier extends far inland. It is not clear whether grounding line retreat into the Fleming Glacier upstream basin will occur without further forcing. Transient simulations with improved knowledge of bed topography are necessary to predict the movement of the grounding line and how long it will take to achieve a new stable state. Coupled ice sheet ocean modelling will be required to explore the evolution of the ice shelf melting and impact of buttressing from the remaining and new ice shelf on the grounded glacier. Future studies of the dynamic evolution of the Fleming Glacier system will enhance our understanding of its vulnerability to marine ice sheet instability and provide projections of its future behavior.

## Author Contribution

Chen Zhao collected the datasets, ran the simulation, and drafted the paper. All authors contributed to the refinement of the experiments, the interpretation of the results and the final manuscript.

## Acknowledgements

Chen Zhao is a recipient of an Australian Government Research Training Program Scholarship and Quantitative Antarctic Science Program Top-up Scholarship. Rupert Gladstone is funded by the European Union Seventh Framework Program (FP7/2007-2013) under grant agreement number 299035 and by Academy of Finland grant number 286587. Matt A. King is a recipient of an Australian Research Council Future Fellowship (project number FT110100207) and is supported by the Australian Research Council Special Research Initiative for Antarctic Gateway Partnership (Project ID SR140300001). Thomas Zwinger's contribution has been covered by the Academy of Finland grant number 286587. This work was supported by the Australian Government's Business Cooperative Research Centres Programme through the Antarctic Climate and Ecosystems Cooperative Research Centre (ACE CRC). This research was undertaken with the assistance of resources and services from the National Computational Infrastructure (NCI), which is supported by the Australian Government. We thank Alex S. Gardner for providing the velocity dataset for 2015. We thank E. Rignot, J. Mouginot, and B. Scheuchl for making their SAR velocities publically available. We thank Yongmei Gong for advice on the analysis of hydraulic potential. SPOT 5 images and DEMs were provided by the International Polar Year SPIRIT project (Korona et al., 2009), funded by the French Space Agency (CNES). This work is based on data services provided by the UNAVCO Facility with support from the National Science Foundation (NSF) and National Aeronautics and Space Administration (NASA) under NSF Cooperative Agreement No. EAR-0735156. The ASTER L1T data product was retrieved from https://lpdaac.usgs.gov/data_access/data_pool, maintained by the NASA EOSDIS Land Processes Distributed Active Archive Center (LP DAAC) at the USGS/Earth Resources Observation and Science (EROS) Center, Sioux Falls, South Dakota.

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

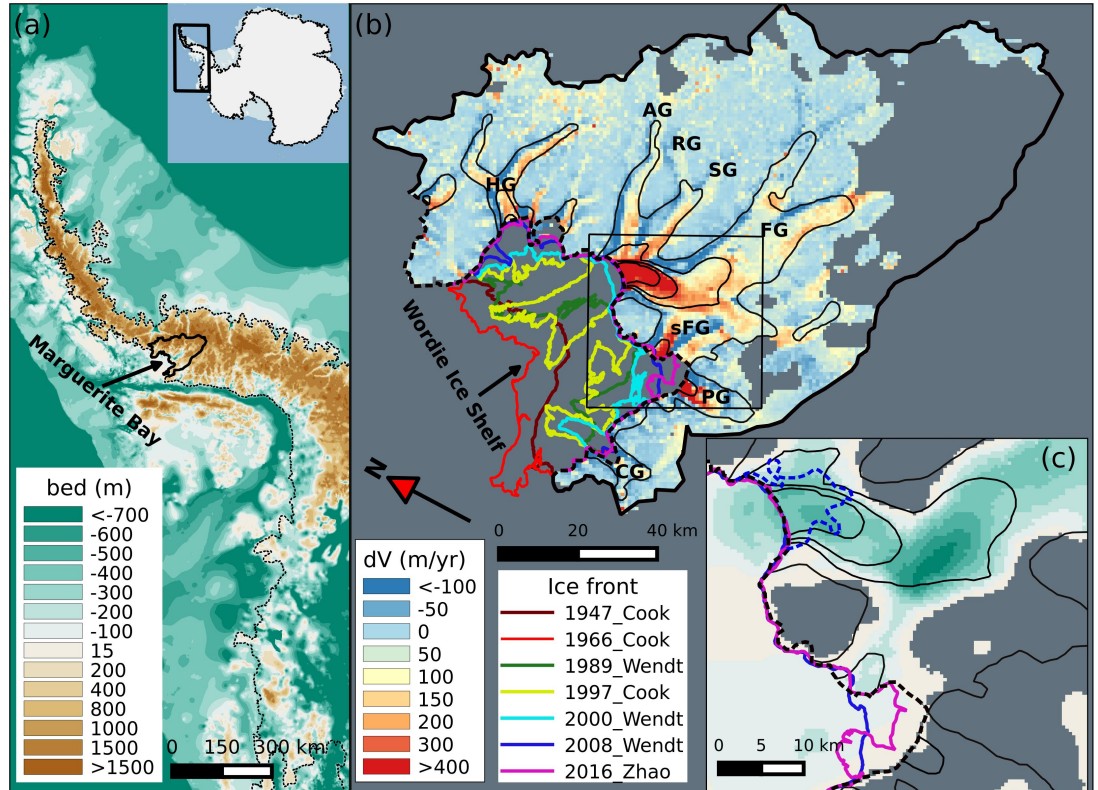

Figure 1. (a) The location of the study region in the Antarctica Peninsula (solid line polygon) with bedrock elevation data "bed_zc" ", based on BEDMAP2 (Fretwell et al., 2013) but refined using a mass conservation method for the fast-flowing regions of the Fleming Glacier system (Zhao et al., companion paper). (b) Velocity changes of the Wordie Ice Shelf-Fleming

Glacier system from 2008 (Rignot et al., 2011c) to 2015 (Gardner et al., 2018). Black contours representing the velocity in 2008 with a spacing of 500 m yr$^{-1}$. The colored lines represent the ice front positions in 1947, 1966, 1989, 1997, 2000, 2008, and 2016 obtained from Cook and Vaughan (2010), Wendt et al. (2010), and Zhao et al. (2017). The feeding glaciers for the Wordie Ice Shelf include three branches: Hariot Glacier (HG) in the north,

Airy Glacier (AG), Rotz Glacier (RG), Seller Glacier (SG), Fleming Glacier (FG), southern branch of the FG (sFG) in the middle, and Prospect Glacier (PG), and Carlson Glacier (CG) in the south. The grey area inside the catchment shows the region without velocity data. (c) Inset map of the Fleming Glacier with ice front positions in 2008 and 2016, grounding line in 1996 (dashed black line) from Rignot et al. (2011a) and deduced grounding line in 2014

(dashed blue line) from Friedl et al. (2018). The background image is the bedrock from panel (a) and the black contours are the same ones as in panel (b).

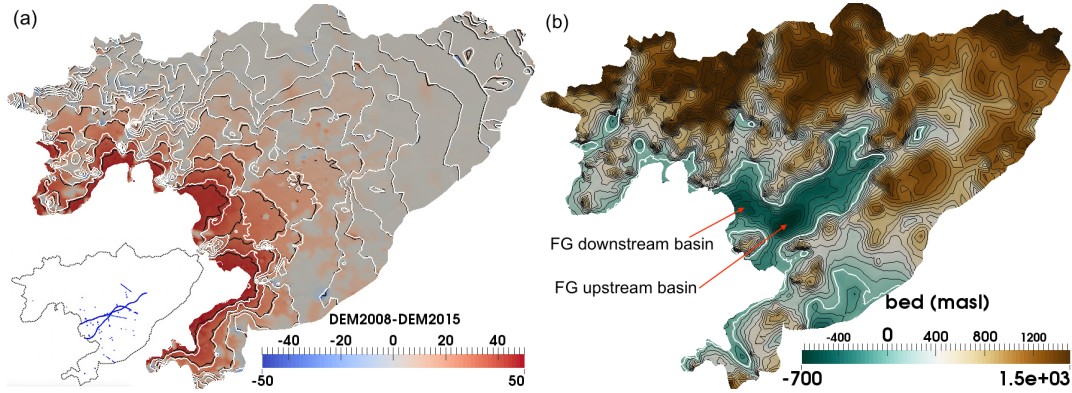

Figure 2. (a) Surface elevation difference between 2008 and 2015 (2008 minus 2015) with black and white contours (interval: 200 m) representing the surface elevation in 2008 and 2015, respectively. Inset map shows the location in the research domain with blue points showing the available elevation data points used to extract the hypsometric model of elevation change from 2008 to 2015 (Zhao et al., 2017). (b) bed elevation data "bed_zc" (metres above sea level, masl) with two basins "FG downstream basin" and "FG upstream basin" from Zhao et al. (companion paper). The black contours show the bed elevation with an interval of 100 m. The white contour represents the sea level used in this study.

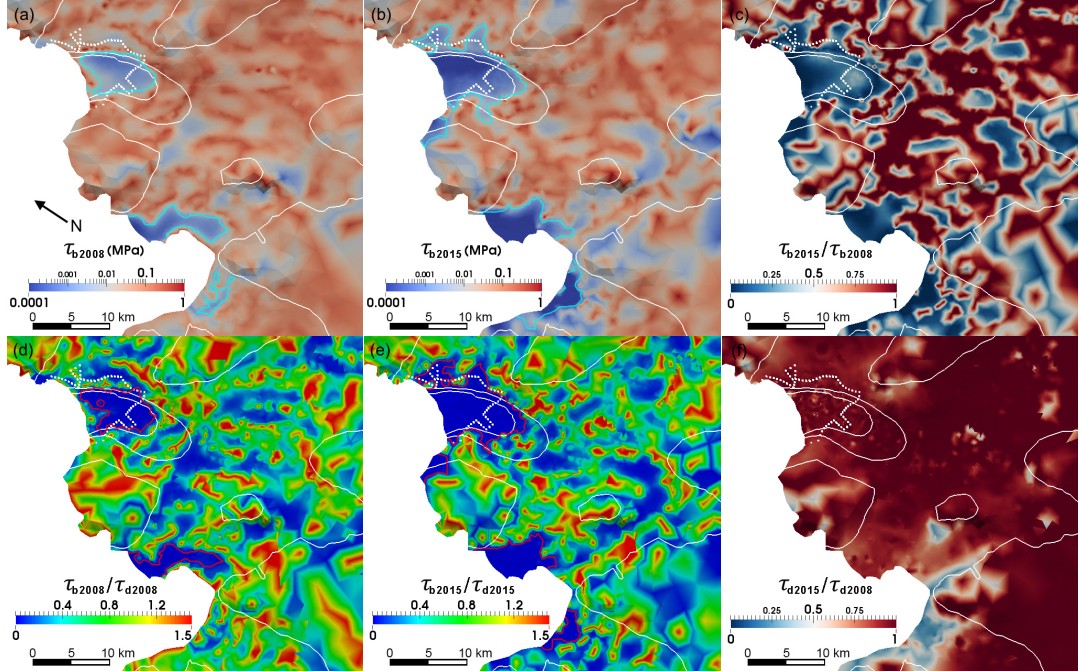

Figure 3. (a,b) Basal shear stress $\tau_b$, (d, e) the ratio of $\tau_b$ to $\tau_d$, of the Fleming Glacier and the Prospect Glacier in 2008 (left) and 2015 (middle). (c) the ratio of basal shear stress $\tau_{b2015}$ to $\tau_{b2008}$, and (f) the ratio of driving stress $\tau_{d2015}$ to $\tau_{d2008}$. The white dotted line represents the deduced grounding line in 2014 from Friedl et al. (2018). The cyan lines in (a) and (b) show the $\tau_b$=0.01 MPa contour. The red lines in (d) and (e) show the RBD = 0.1 contour in the current study. The white solid lines represent the 2008 surface speed contours of 100 m yr$^{-1}$, 1000 m yr$^{-1}$, and 1500 m yr$^{-1}$, respectively, to aid visual comparison across subplots.

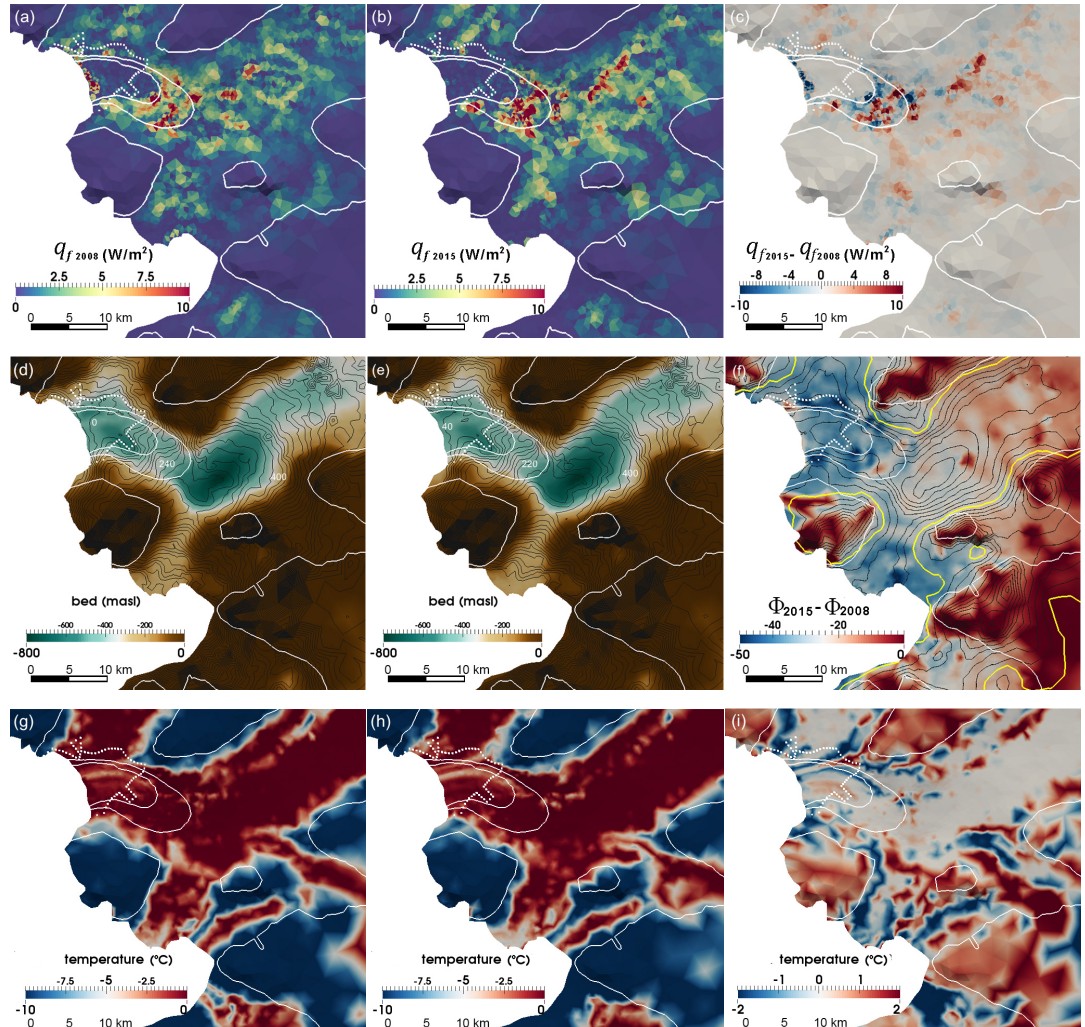

Figure 4. (a, b) The basal friction heating, (d, e) the contours of hydraulic potential with a spacing of 20 m (black solid lines) with the bed elevation (metres above sea level) as the background, and (g, h) the simulated temperature relative to the pressure melting point at the base of the Fleming Glacier and the Prospect Glacier in 2008 (left) and 2015 (middle). The differences of (c) basal friction heating, (f) hydraulic potential, and (i) simulated basal temperature between 2008 and 2015 (2015 minus 2008). The black contours in (f) represent the bedrock elevation with a spacing of 100 m. The white dotted line represents the deduced grounding line in 2014 from Friedl et al. (2018). The white solid lines represent the 2008 surface speed contours of 100 m yr$^{-1}$, 1000 m yr$^{-1}$, and 1500 m yr$^{-1}$.

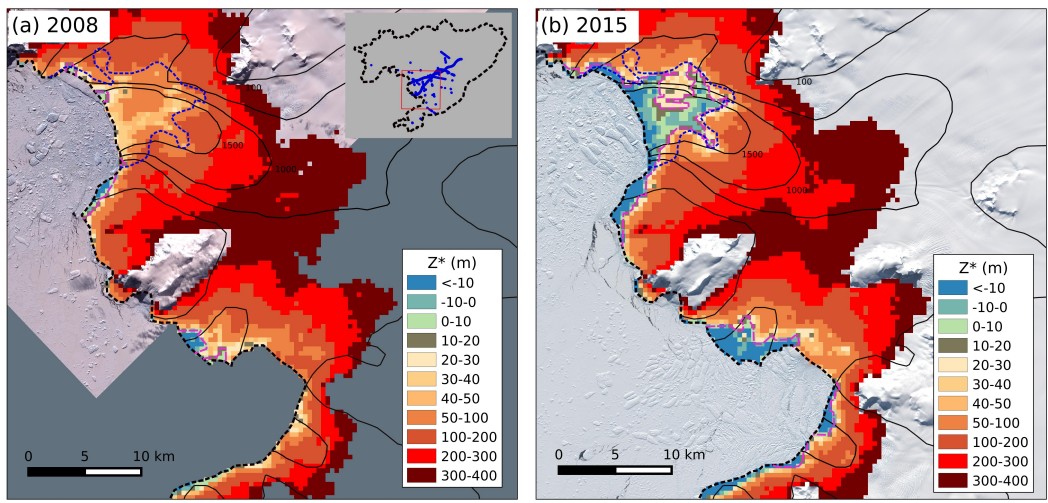

Figure 5. The height above buoyancy $Z_*$ in (a) 2008 and (b) 2015 of the Fleming Glacier and Prospect Glacier. The background images are from (a) ASTER L1T data in Feb 2[nd], 2009, and (b) Landsat-8 in Jan 13[th] 2016, respectively. The black lines represent velocity contours in 2008 (Rignot et al., 2011c). The dashed black and blue lines show the grounding line in 1996 (Rignot et al., 2011a) and 2014 (Friedl et al., 2018), respectively. The dashed magenta line shows the possible grounding line with $Z_* < 20$ m. Inset map shows the location in the research domain with blue points showing the available elevation data points used to extract the hypsometric model of elevation change from 2008 to 2015 (Zhao et al., 2017).