# Peer review of "Basal friction of Fleming Glacier, Antarctica, Part B: evolution from 2008 to 2015"

_The Cryosphere, 2017_

## Referee Comment (RC1) · Anonymous Referee #1 · 5 Feb 2018

General comments

This paper presents some interesting results suggesting that glacier-bed interactions have an important role in the dramatic speedup of Fleming Glacier, Antarctica from 2008-2015. Recent work by Walker and Gardner posited that abnormally warm ocean temperatures in Marguerite Bay over this time period caused the observed changes in the glaciers that fed the former Wordie Ice Shelf. Friedl et al. 2017 used a combination of several remote sensing data sets to show that large areas near the terminus of Fleming Glacier ungrounded between 2008-2015. These data sets showed that Fleming Glacier lies on a retrograde bed slope, and thus is susceptible to runaway retreat via marine ice sheet instability. Zhao et al. argue that both of these explanations leave out a key factor, the interaction of the glacier and its bed. The authors used inverse

methods to estimate the basal shear stress under Fleming Glacier in 2008 and 2015. This analysis revealed a band of high basal shear stress near the terminus in 2008 that is no longer present in 2015. They argue that the retreat of the glacier off of this region of high basal friction may also be a factor in the subsequent speedup.

The large changes that Fleming Glacier exhibited make it a valuable test case for understanding glacier change in Antarctica. The authors' results suggest that glacier-bed interactions are an important factor, in addition to ocean melting and geometric instability, in understanding the recent behavior of the glaciers that fed the former Wordie Ice Shelf. While I recommend publication, the work could be improved on a number of fronts. Drawing conclusions about the physics of glacier-bed interactions from the results of inverse methods can be difficult because, as the authors acknowledge, any one feature could be an artifact. Many of the arguments made in the paper are speculative and I think this should be made clearer. Suppose that the high-basal shear band they claim to find at the terminus of Fleming Glacier in 2008 were merely an artifact – what would that mean for the physics?

Specific comments

First, I think the abstract could be improved by (1) cutting many of the details that are covered in the discussion section and (2) giving a clearer statement about what this paper adds to the existing knowledge. The main precedents that the authors draw from are Walker and Gardner 2017 and Friedl et al. 2017. What do these two papers conclude, and how do the authors' conclusions agree with or depart from them? For example, both the present work and Friedl et al. 2017 argue that the speedup and thinning of Fleming Glacier is a consequence of ongoing marine ice sheet instability. However, the authors argue that subglacial hydrological effects may have also initiated the retreat off of a stable bedrock high, while Friedl et al. point solely to ocean warming. To my knowledge, this work and the companion paper are the first to use inverse methods to estimate the basal shear stress of this particular site at high resolution, as opposed to low-resolution estimations for all of Antarctica. This information, which

ideally would be front-and-center in the abstract, is partly obscured by details that will be addressed in the discussion section anyway. In any case, this problem is more one of presentation and not of actual content.

At several points, the authors pose the question of whether either ocean warming or basal processes are the "dominant" causes of the observed changes (see lines 84, 411). The question of which process is dominant assumes that the two are additive, but if instead the relationship is causative, this question ceases to be meaningful. In the discussion section, the authors suggest that hydrological effects could destabilize the high-friction band, resulting in speedup, thinning, and ungrounding. In this scenario, hydrology-induced speedup and ungrounding create conditions where the ocean can then melt the ice shelf from underneath. One could also imagine a scenario in which ocean melting comes first and hydrological effects second. For example, ocean melting could push the glacier terminus off of a highly resistive bedrock bump, and the glacier begins to speed up and thin. The reduced overburden pressure then changes the overall hydraulic potential. The authors' hypothesis that hydrology might have initiated the recent changes is still significant and worth considering. Nonetheless, the paper's intent might be clearer by changing questions about which process is dominant to questions about which one came first. Finally, the authors suggest that coupled ice sheet-ocean modeling could help determine which case is more likely. This point could be expanded on further. For example, a coupled model using pre-2006 values of ocean heat flux that does exhibit a hydrology-induced destabilization would show that oceanic forcing is not necessary to explain observations.

The text gives conflicting statements about the authors' degree of confidence in the veracity of their conclusions. For example, in line 314 the authors assert that the disappearance of the high friction band near the calving front is a "likely" trigger for the subsequent retreat, but at other points they equivocate about whether this feature is real or merely an artifact. A lack of complete certainty about this resistive band is entirely reasonable but the paper would be improved if it were more consistent in what

kind of assertions are made.

A numerical experiment could shed some more light on whether the resistive band near the terminus is real or not. The methods section describes inferring the basal friction using the 2008 ice thickness and the 2015 velocity to examine whether the result is sensitive to the geometry. In this vein, the authors could compute a velocity using the 2015 basal friction and the 2008 thickness. How well does this computed velocity agree with observations, weighted by the error variances? Is the misfit worse than that of the velocity computed using both the 2008 thickness and basal friction? If so, by how much? The presence of a resistive band at the terminus would be doubtful if a basal friction field without this feature can explain the 2008 data just as well as a basal friction field with this feature.

Technical corrections

17-21: Flip the order of the sentences starting with "To explore the mechanism underlying these changes..." and "Recent observational studies..."

23-28: Giving too much justification in the abstract obscures your overall point, this could be moved to the discussion.

66-69: "As a marine-type glacier system..." Rephrase or break up into 2 sentences.

73-74: "None of these past studies have modelled the glacier system and hence these hypotheses are untested." This suggests that modelling is the only way to really test these hypothesis. It's better to just say that the precise nature of the feedbacks hasn't been established and that you will test them using models.

88-90: "Changes in basal shear stress..." Rephrase this sentence.

162-165: "To explore their relative impacts..." While this experiment is a good sanity check, the result isn't essential to making your point and this could be relegated to a supplement.

294-296: Make this "The change in area", and "additional evidence supporting the hypothesis of rapid grounding line retreat".

313-315: Could the basal resistance band at the front be an artifact of neglecting back-stress at the terminus from melange or sea ice?

324-327: Overly long sentence, break up into 2 sentences.

326-327: "...as in the rapid retreat of Jakobshavn Isbrae in West Greenland (Steiger et al., 2017)." There were other factors in the retreat of Jakobshavn, see Motyka et al. 2011 and Holland et al. 2008.

336-340: Run-on sentence, break up into 2 or 3 sentences.

400-402: "...hard to say how much forcing would be needed to push the grounding line into it." Rephrase.

414: Change "simulate" to "estimate".

---

## Referee Comment (RC2) · Anonymous Referee #2 · 6 Mar 2018

**The Cryosphere TC2017-242** *"Basal drag of Fleming Glacier, Antarctica, Part B: implications of evolution from 2008 to 2015"* by Zhao and others.

This paper, using diagnostic inverse modeling of basal conditions, discusses the possible causes of the retreat of Fleming glacier observed between 2008 to 2015. In particular, the potential acceleration induced by the production of water by frictional heating at the base of the glacier is discussed. This paper is well written, even if some sentences are too long and some figures can be improved. I have made below some suggestions that I believe could improve the manuscript.

[Figure]

line 62: nearly twice or more than twice?

line 95: I don't really see where in Gladstone et al. (2017) inverse methods are used?

line 123: define what is bed_zc

line 127: $S_{2008}$ is not the "surface DEM in 2008" but the "surface elevation in 2008".

line 134: (on the same line) The "2008 velocity" should be "The 2008 velocity dataset"

line 155: the assumption that all the ice is grounded is for the inverse method? May be you can specify already here that floating ice will be deduced as the place where basal stress is lower than a threshold? It is not clear all along the manuscript if there is still a floating part or not on Fleming glacier and it would help if it could be mentioned more clearly in the introduction.

line 175: it should be mentioned that Eq. (4) is valid under the assumption of $N = 0$

line 186: here it should be mentioned that Eq. (6) is derived under the assumption of a perfect connectivity of the basal hydrology system with the ocean

line 192: $C$ is not a vector (not in bold)

line 380: The increase of the amount of melt water should be quantified by integrating the frictional heating over the bedrock. But it should be also discussed that more melt doesn't necessarily induce an acceleration of the glacier as the basal hydrology system is evolving dynamically to adjust this surplus of water. The link of basal sliding with basal water should be clarified, and specifically is

should be mentioned that the important variable is not the amount of water but its pressure. And this later quantity is not evaluated in the present work.

line 430: Can the buttressing exerted by the pining band in 2008 be quantified in a more rigorous way? A complementary experience would be to remove this band of high friction (by setting no friction there) and to see how the velocity field is modified upstream. This would directly quantify the increase of velocity induced by an instantaneous loss of the pining band. The difference between this velocity field and the 2015 one would indicate places where a decrease of basal shear stress is necessary to explain the 2015 velocity field.

line 528: Schaëfer is not spelled correctly

Âăcaption Fig. 1: inset (c) should be located in (b) and in (c) the front position in 2008 and 2016 should be added to visualise a potential ice-shelf?

Fig. 3: the grounding line in 2014 seems to have a different form than the one of Friedl et al. (2017) in their Fig. 6?

---

## Referee Comment (RC3) · Anonymous Referee #3 · 12 Mar 2018

GENERAL COMMENTS

Main question in the abstract: Is the observed acceleration of the flow and thinning of the glacier due to increased ocean warming and/or marine ice sheet instability?

Method: Infer basal shear stress from observations and calculate a steady state temperature field using a Stokes ice sheet model for 2008 and 2015.

Results: Reduction in magnitude and increase in area of low basal shear stress near the 1996 grounding line and reduction in height above flotation between 2008 and 2015 suggest the grounding line has retreated for Fleming Glacier, southern branch of Fleming Glacier and Prospect Glacier.

A band of higher basal shear stress parallel to the 1996 grounding line at 2008 suggests that Fleming Glacier was still grounded at that time.

Subglacial water may be generated from high basal frictional heating upstream of Fleming Glacier. Frictional heating has increased between 2008 and 2015 over a rise between two deep bedrock basins.

Comments: I don't think the main question can be answered from instantaneous time slices of the ice flow. The authors need to do forward experiments with various ocean forcing such as different basal melt rates or vertical melting at the calving front. Alternatively, the authors need to pose a different question.

The band of high basal shear stress may not be physical realistic. The model error reported in their companion paper is relatively high in this area.

Interesting idea: The authors propose that basal water generated from high basal frictional heating upstream draining towards the front, triggered grounding line retreat of Fleming Glacier. This mechanism is an alternative to the usual ocean forcing explanation. Mass loss could significantly increase, due to marine instability, if the grounding line retreated over a bedrock rise into the second deeper basin. The highest frictional basal heating in 2015 is located over the rise, which may be a potential trigger for the grounding line retreat.

Manuscript in general: The font is too small and the text is not double spaced, which made reviewing the paper tricky. Picking out the references was particularly difficult give the font size and text spacing. Some of figures are too small.

SPECIFIC COMMENTS

Ocean forcing: It seems reasonable to suggest that increased melting at the vertical face of the front of FGL due to incursions of CDW may have affected the pressure boundary condition at the front sufficiently to remove the high band of basal shear stress. However, I don't think your results shed any new light on what has been suggested in the other references you use about ocean basal melting. Forward timedependent modelling experiments are needed to test these theories and here's an example for Larsen B of how you can extend the work you have done for this paper. Vieli et al 2007 Causes of pre-collapse changes of the Larsen B ice shelf: Numerical modelling and assimilation of satellite observations. Earth and Planetary Science Letters. https://doi.org/10.1016/j.epsl.2007.04.050

Grounding line retreat: The results for 2015 of low basal shear stress and low height above buoyancy confirm the findings of Friedl et al 2017 that Fleming Glacier's grounding line has retreated. The results for PGL are different to FGL: Driving stress appears to be much higher for PGL in 2015. Temperature homologous near the 1996 grounding line appears much lower in 2015 suggesting that the glacier may have become frozen to the bed?

Is the band of high basal stress at the front of FGL physically realistic? The authors attempt to address this question in the paragraph beginning on line 209. Part A shows that the misfit between the modelled and observed speed is high, where the modelled speed is too fast, and the surface slope is also higher here than over the region of low shear stress. The driving stress is not obviously high given the relatively high surface slope. What concerns me is your model appears unable to model the front. What about rheology of the ice near the front? Perhaps the standard A is not appropriate here. Part A shows a large vertical shear at the front where the basal speed is much smaller than the surface. Is the ice stiffer at the front? Vieli et al 2006 Numerical modelling and data assimilation of the Larsen B ice shelf, Antarctic Peninsula, Phil. Trans. R. Soc. A, 364, 1815–1839, doi:10.1098/rsta.2006.1800 solved the inversion problem for effective viscosity. Modelling a front is difficult! What about the direction of the flow? Is there a difference is the modelled flow direction and the observed direction? Is there a change in flow direction between 2008 and 2015 as the ice moves over the sticky band and becomes ungrounded. Also, could ice melange at the front FGL affect the boundary condition?

Basal frictional heating and subglacial water: Could the region of low basal shear stress

near the front simply be due to subglacial water from upstream pooling in the bedrock basin, e.g. FGL in 2008? Could the region be partially grounded? The temperature homologous is high, which prevents the water from refreezing. I think the role of subglacial water could be explained more in the literature review. Paragraph beginning 85: You don't explain what the feedback mechanism is. As I understand it, Schoof (2010) talks about the importance of variability of basal water on flow dynamics, with flow accelerating due to a short-lived increase in basal water, but then the flow slows if the basal water stays high. Is that happening here? You have high basal frictional melting in 2015, which you say is speeding up the flow, but figure 3 in Friedl et al 2017 shows that the ice speed of FGL decreased between 2011-2015.

Figure 1 shows that the ice front for PGL and sFGL has calved between 2008 and 2016, with some advance in the southern part of the bay. Could calving event(s) explain the speed up and lowering of the surface of the streams? Also, from figure 1, PGL has an ice shelf, are you applying the normal stress, hydrostatic pressure boundary condition at the 1996 grounding line or at the real front?

Bedrock plots: The way the bedrock is plotted is a bit inconsistent and unclear. Figure 1 c clearly shows where bedrock is above or below sea level with a white colour band around 0, but the bed elevation colour in figure 4 c and d looks like most of the bedrock is either below or at sea level and figure 2 b is too small. It would be useful to see where the retrograde and prograde slope are.

Figures 3,4,5: useful to have a third column of figures showing the difference between the first two columns.

Line 158: The Linear sliding law is fine for the inverse problem because $\tau$ remains unchanged, if a higher power of u is used, only the coefficient C would change. However, for a forward run a linear law may be inappropriate.

Please define basal frictional heating in your method section.

Figure 2 is too small and/or too detailed. A difference plot of surface elevation may be more informative.

Figure 3: The patterns in c and d seem to be influenced by the computational mesh. Have you investigated mesh resolution by halving or doubling element sizes?

The ratio of basal shear stress to driving stress: I'm not sure what figure 3 c and d are showing or why a low value means the ice may be close to flotation. Figure 4 c and d: Would showing the potential gradient be more informative? I can't see labels on the contours of hydraulic potential.

Figure 5: Height above buoyancy appears to be negative south of the main stream of FGL (and south of PGL). Is the bedrock above sea level there?

Discussion section: Is a maximum melt rate of 1 m/a enough to generate a plume of high enough velocity to entrain incursions of CDW to enhance basal melting beneath the floating ice? You can calculate the flux of subglacial water for each year by $\tau$bub/Lii x area that feeds the grounding line based on the hydraulic potential (or its gradient).

Line 395: Could you explain the positive feedbacks.

Estimating the time scale for the ice to unground from the rise between basins leaving the ice stream vulnerable of marine instability in the upstream basin is good, but I'm not sure you can say height above buoyancy is a measure of potential mass loss.

TECHNICAL CORRECTIONS

Line 46: abbreviation 'GL' is not defined.

Line 88: Not sure the sentence is helpful. Might be better to delete it.

Section 2.2: Is Hmc part of a dataset from Morlighem or have you combined two dataset yourself?

Lines 132, 151: Part 1 or Part A

Line 145: Is the basal frictional heating calculated from output from the inverse problem and used as an input into the heat equation?

184: I don't think N needs a numbered equation because it isn't used.

Line 222: northern and eastern. It might be helpful to add an arrow indicating North on one of the figures.

Lines 367, 420: Friedl et al 2017 gave an estimated grounding line for 2014.

Figure 1: sFGL is not marked on the figure.

Figure 3: It is difficult to work out where the plotted regions exist in relation to figures 1 and 2. Orientation is given in figure 5 but would be more useful on figure 3.

Figure 3: Cannot see cyan contour on printed paper.

Figure 4: I can't distinguish between red and magenta contours.

Line 529: Case is wrong for Schafer.

---

## Author Comment (AC1) · 19 May 2018

We are grateful to Reviewer 1 for the positive and constructive suggestions to improve our paper. We have addressed the comments below. The line numbers in the responses are based on the revised manuscript without change track.

Please note that Mathieu Morlighem created the ice thickness data for the Fleming Glacier system using the mass conservation method, which is very important for most experiments done in this study. We do value his contribution to this paper, so we add him as the co-author in the revised text.

In the revised companion paper (Zhao et al., companion paper), we implemented a new sensitivity test to the enhancement factor (E). It reveals that the optimal value of $E = 1.0$ should be chosen as the enhancement factor in the CONTROL experiment. Accordingly, we re-ran all the simulations in this study with $E = 1.0$, and the high basal shear stress band near the ice front in 2008 has decreased into high basal shear spots, which are suspected of being artefacts of the inversion process and are discussed below. We modified the text and figures accordingly. All other result and interpretations are not qualitatively changed from the original manuscript.

General comments

This paper presents some interesting results suggesting that glacier-bed interactions have an important role in the dramatic speedup of Fleming Glacier, Antarctica from 2008-2015. Recent work by Walker and Gardner posited that abnormally warm ocean temperatures in Marguerite Bay over this time period caused the observed changes in the glaciers that fed the former Wordie Ice Shelf. Friedl et al. 2017 used a combination of several remote sensing data sets to show that large areas near the terminus of Fleming Glacier ungrounded between 2008-2015. These data sets showed that Fleming Glacier lies on a retrograde bed slope, and thus is susceptible to runaway retreat via marine ice sheet instability. Zhao et al. argue that both of these explanations leave out a key factor, the interaction of the glacier and its bed. The authors used inverse methods to estimate the basal shear stress under Fleming Glacier in 2008 and 2015. This analysis revealed a band of high basal shear stress near the terminus in 2008 that is no longer present in 2015. They argue that the retreat of the glacier off of this region of high basal friction may also be a factor in the subsequent speedup.

The large changes that Fleming Glacier exhibited make it a valuable test case for understanding glacier change in Antarctica. The authors' results suggest that glacier-bed interactions are an important factor, in addition to ocean melting and geometric instability, in understanding the recent behavior of the glaciers that fed the former Wordie Ice Shelf. While I recommend publication, the work could be improved on a number of fronts. Drawing conclusions about the physics of glacier-bed interactions

from the results of inverse methods can be difficult because, as the authors acknowledge, any one feature could be an artefact. Many of the arguments made in the paper are speculative and I think this should be made clearer. Suppose that the high-basal shear band they claim to find at the terminus of Fleming Glacier in 2008 were merely an artefact – what would that mean for the physics?

This is a great suggestion. As mentioned above, in the modified companion paper (Zhao et al., companion paper), we speculate that the high basal shear spots near the ice front may be artefacts. However, the possibility of the high basal friction spots being real features, which might be caused by pinning points near the 1996 grounding line position is not excluded. Based on the inferred basal shear stress (Fig. 3a) and height above buoyancy (Fig. 5a), the 1996 grounding line position may have not retreated prior to Jan 2008, and Friedl et al. (2018) also suggested that the grounding line position may have retreated behind the 1996 position after Jan-Apr 2008.

The discussion in the manuscript has been modified to respond to the reviewer's comment adding (Line 361-365): "If the high basal resistance spots are artefacts, ungrounding of this region in early 2008 is less viable as an explanation for an abrupt increase in ice flow speed, since the loss of backstress would be more gradual. In this case, positive feedbacks, such as the marine ice sheet instability or the basal melt feedback, are even more likely to explain the FG's recent behavior."

In general, with regard to inverse methods, small features can more easily arise as inversion artefacts than larger features. Small basal shear stress features may be locally balanced by extensional/compressional stresses in the ice without needing to balance the gravitational driving stress. For features with larger horizontal scales basal shear stress must approximately balance the driving stress and these features are less likely to be artefacts. All features discussed in the paper arising from the inversion process, aside from the sticky spots near the 2008 front, are large enough that we are confident they are robust features of the inversion and not artefacts.

Currently one possibility for rapid retreat of the grounding line is that there were some sticky spots near the front, and rapid retreat occurred when the ice ungrounded from these sticky spots.

Specific comments

First, I think the abstract could be improved by (1) cutting many of the details that are covered in the discussion section and (2) giving a clearer statement about what this paper adds to the existing knowledge. The main precedents that the authors draw from are Walker and Gardner 2017 and Friedl et al. 2017. What do these two papers conclude, and how do the authors' conclusions agree with or depart from them? For example, both the present work and Friedl et al. 2017 argue that the speedup and thinning of Fleming Glacier is a consequence of ongoing marine ice sheet instability. However, the authors argue that subglacial hydrological effects may have also initiated the retreat off of a stable bedrock high, while Friedl et al. point solely to ocean warming. To my knowledge, this work and the companion paper are the first to use inverse methods to estimate the basal shear stress of this particular site at high resolution, as opposed to low-resolution estimations for all of Antarctica. This information, which ideally would be front-and-center in the abstract, is partly obscured by details that will be addressed in the discussion section anyway. In any case, this problem is more one of presentation and not of actual content.

Thanks for the reviewer's suggestion. We added this point in the first paragraph of

Section 4.1 (Line 225-229) "Although low-resolution estimation of basal shear stress has been carried out for the whole Antarctic Ice Sheet (Fürst et al., 2015; Morlighem et al., 2013; Sergienko et al., 2014), this is the first application of inverse methods to estimate the basal friction pattern of the Fleming system at a high resolution and use the full-Stokes equations." and modified the conclusion and abstract correspondingly.

At several points, the authors pose the question of whether either ocean warming or basal processes are the "dominant" causes of the observed changes (see lines 84, 411). The question of which process is dominant assumes that the two are additive, but if instead the relationship is causative, this question ceases to be meaningful.

The reviewer seems to think that when we say "dominant" we mean "only". Of course, it is possible to have a small perturbation caused by one process (ocean-warming driven basal melting) and massively enhanced by another process (basal process). In this case we would describe the latter as "dominant" because it has caused the biggest change, even if there would have been no change without the former.

In the discussion section, the authors suggest that hydrological effects could destabilize the high-friction band, resulting in speedup, thinning, and ungrounding. In this scenario, hydrology-induced speedup and ungrounding create conditions where the ocean can then melt the ice shelf from underneath. One could also imagine a scenario in which ocean melting comes first and hydrological effects second. For example, ocean melting could push the glacier terminus off of a highly resistive bedrock bump, and the glacier begins to speed up and thin. The reduced overburden pressure then changes the overall hydraulic potential. The authors' hypothesis that hydrology might have initiated the recent changes is still significant and worth considering. Nonetheless, the paper's intent might be clearer by changing questions about which process is dominant to questions about which one came first. Finally, the authors suggest that coupled ice sheet-ocean modeling could help determine which case is more likely. This point could be expanded on further. For example, a coupled model using pre-2006 values of ocean heat flux that does exhibit a hydrology-induced destabilization would show that oceanic forcing is not necessary to explain observations.

We don't see a need to choose to only consider which process comes first or which process is dominant. The two questions are complementary rather than contradictory. When we discuss which process is dominant we do not mean to exclude the relevance of which came first. It was not our intention to propose that the changes were initiated by the subglacial hydrologic system – we don't have a mechanism in mind for that. We don't see how an increase in subglacial melting can happen without an external trigger, except through increased insulation due to ice thickening such as occurs in surging glaciers. But we doubt this is happening here. We suspect the ice shelf collapse triggers a positive feedback at the bed of the fast flow region, and that once the shelf has gone, the melt rates due to the ocean warming do not make much difference. Subglacial melting probably has to be happening all the time under the fast flowing region in any case. Ocean melting/ice shelf collapse provide a triggering mechanism to the ungrounding process, and then the positive feedback between the basal sliding and subglacial water pressure at the bed kicks in. We have clarified the nature and role of this positive feedback mechanism in the Sect. 4.2 and Sect. 5.

We don't think there is a need to expand further about designing coupled experiments as that is well outside the scope of this paper.

The text gives conflicting statements about the authors' degree of confidence in the veracity of their conclusions. For example, in line 314 the authors assert that the disappearance of the high friction band near the calving front is a "likely" trigger for the subsequent retreat, but at other points they equivocate about whether this feature is real or merely an artefact. A lack of complete certainty about this resistive band is entirely reasonable but the paper would be improved if it were more consistent in what kind of assertions are made.

Thanks for pointing this out. Based on the modified companion paper (Zhao et al., companion paper), we speculate that the high basal shear spots in 2008 (rather than the band of high shear seen in the previous version) may be artefacts but we do not rule out the possibility of high friction spots as a real feature caused by the pinning points at the 1996 grounding line. For consistency we modified the text in the manuscript accordingly (Line 359-361).

For the example mentioned by the reviewer, we modified "likely" to "possible" (Line 383). Under this speculation, if the sticky spots were totally artefacts, the reduction in basal drag would be likely due to the positive feedbacks between the basal sliding and basal subglacial water.

A numerical experiment could shed some more light on whether the resistive band near the terminus is real or not. The methods section describes inferring the basal friction using the 2008 ice thickness and the 2015 velocity to examine whether the result is sensitive to the geometry. In this vein, the authors could compute a velocity using the 2015 basal friction and the 2008 thickness. How well does this computed velocity agree with observations, weighted by the error variances? Is the misfit worse than that of the velocity computed using both the 2008 thickness and basal friction? If so, by how much? The presence of a resistive band at the terminus would be doubtful if a basal friction field without this feature can explain the 2008 data just as well as a basal friction field with this feature.

The basal friction field without the sticky spots cannot explain the 2008 data. Although we are not sure whether the high basal drag spots in 2008 are real or not, we are sure that the basal drag of high velocity regions in 2008 should not be as small as that in 2015. However, we still tried this experiment as the reviewer suggested. Results show that the simulated surface velocity was nearly 2.5 times the observed surface velocity of 2008 near the ice front. So we cannot use the suggested experiment to say that the sticky spots are an artefact.

Technical corrections

17-21: Flip the order of the sentences starting with "To explore the mechanism underlying these changes..." and "Recent observational studies..."

Modified.

23-28: Giving too much justification in the abstract obscures your overall point, this could be moved to the discussion.

We agree to remove the sentence about the grounding line position in 2008, but we think the comparison results between 2008 and 2015 should appear in the Abstract.

66-69: "As a marine-type glacier system..." Rephrase or break up into 2 sentences.

The whole sentence has been modified into "As a marine-type glacier system residing

on a retrograde bed with bedrock elevation as much as ~800 m below sea level (Fig. 1c), the Fleming system is hence potentially vulnerable to marine ice sheet instability (Mercer, 1978; Thomas and Bentley, 1978; Weertman, 1974). The acceleration and greater dynamic thinning of the FG over 2008-2015 suggests the possible onset of unstable rapid grounding line retreat (Walker and Gardner, 2017; Zhao et al., 2017), which has been confirmed by Friedl et al. (2018). " (Line 74-79).

73-74: "None of these past studies have modelled the glacier system and hence these hypotheses are untested." This suggests that modelling is the only way to really test these hypothesis. It's better to just say that the precise nature of the feedbacks hasn't been established and that you will test them using models.

Thanks for pointing this out. We modified this sentence into "An alternative hypothesis is that the recent changes arise from feedbacks in the dynamics of the evolving glacier, possibly involving the subglacial hydrology. The examination of changes in basal shear stress distributions between 2008 and 2015 in this modelling study provides a first step in exploring possible feedback hypotheses. " (Line 83-87).

88-90: "Changes in basal shear stress..." Rephrase this sentence.

Reviewer 3 has suggested deleting this sentence, since it is not helpful here. We agree with Reviewer 3, so we delete this sentence.

162-165: "To explore their relative impacts..." While this experiment is a good sanity check, the result isn't essential to making your point and this could be relegated to a supplement.

Thanks for the suggestion. We have moved this part into the Sect. S1 in the supplementary material.

294-296: Make this "The change in area", and "additional evidence supporting the hypothesis of rapid grounding line retreat".

We modified it into "This change in area" and "additional evidence supporting the hypothesis of rapid grounding line retreat" (Line 354-356).

313-315: Could the basal resistance band at the front be an artefact of neglecting backstress at the terminus from melange or sea ice?
We have discussed this in Sect. 4.4 of the revised companion paper (Zhao et al., companion paper). The ice mélange back force (~1.1e7 N m$^{-1}$) used to prevent the rotation of an iceberg at the calving front (Krug et al., 2015) could account for the equivalent of up to ~2.3 m sea level in terms of ice front boundary condition. The experiment with the sea level increased by 10 m shows that the high basal shear spots are decreasing but have not disappeared. The situation at the front is complicated. Sea level, bedrock/ice thickness uncertainty, mélange backstress, ice front positions, these things can all impact on our inversion near the ice front.

324-327: Overly long sentence, break up into 2 sentences.

Modified into "For a glacier lying on a retrograde slope in a deep trough, the grounding line may be vulnerable to rapid retreat without any further change in external forcing, once its geometry crosses a critical threshold, which is the marine ice sheet instability hypothesis (e.g., Mercer (1978); Thomas and Bentley (1978); Weertman (1974)). A similar theory has been proposed on the prospective rapid retreat of Jakobshavn Isbræ in West Greenland without any trigger after detaching from a pinning point (Steiger et al., 2017)." (Line 394-399).

326-327: "...as in the rapid retreat of Jakobshavn Isbrae in West Greenland (Steiger et al., 2017)." There were other factors in the retreat of Jakobshavn, see Motyka et al. 2011 and Holland et al. 2008.

Yes, we agree with the reviewer. We should have made it clear that we were talking about the future behavior of the Jakobshavn here. Steiger et al., 2017 found that after decades of stability and with constant external forcing, the grounding lines of Jakobshavn may retreat rapidly without any trigger due to losing the pinning-points. To make it clearer, we modified it into "A similar theory has been proposed on the prospective rapid retreat of Jakobshavn Isbræ in West Greenland without any trigger after detaching from a pinning point (Steiger et al., 2017)."

336-340: Run-on sentence, break up into 2 or 3 sentences.

Modified into "If the system remains out of balance and continues to thin, the grounding line could eventually move across this bed obstacle. If this occurs, the grounding line is then likely to retreat rapidly down the retrograde face of the FG upstream basin, likely to be accompanied by further glacier speed up and dynamic thinning. " (Line 409-413)

400-402: "...hard to say how much forcing would be needed to push the grounding line into it." Rephrase.

Modified into "More thinning would be needed to destabilise the upstream basin, and it is hard to estimate how much forcing would be needed to push the grounding line into the upstream basin boundary." (Line 497-499).

414: Change "simulate" to "estimate".

Modified.

References
Friedl, P., Seehaus, T. C., Wendt, A., Braun, M. H., and Höppner, K.: Recent dynamic changes on Fleming Glacier after the disintegration of Wordie Ice Shelf, Antarctic Peninsula, The Cryosphere, 12, 1-19, 2018.
Krug, J., Durand, G., Gagliardini, O., and Weiss, J.: Modelling the impact of submarine frontal melting and ice mélange on glacier dynamics, The Cryosphere, 9, 989-1003, 2015.
Mercer, J. H.: West Antarctic ice sheet and CO2 greenhouse effect: a threat of disaster, Nature, 271, 321, 1978.
Steiger, N., Nisancioglu, K. H., Åkesson, H., de Fleurian, B., and Nick, F. M.: Non-linear retreat of Jakobshavn Isbræ since the Little Ice Age controlled by geometry, The Cryosphere Discuss., 2017, 1-27, 2017.
Thomas, R. H. and Bentley, C. R.: A model for Holocene retreat of the West Antarctic Ice Sheet, Quaternary Research, 10, 150-170, 1978.
Walker, C. C. and Gardner, A. S.: Rapid drawdown of Antarctica's Wordie Ice Shelf glaciers in response to ENSO/Southern Annular Mode-driven warming in the Southern Ocean, Earth and Planetary Science Letters, 476, 100-110, 2017.
Weertman, J.: Stability of the Junction of an Ice Sheet and an Ice Shelf, Journal of Glaciology, 13, 3-11, 1974.
Zhao, C., Gladstone, R., Zwinger, T., Warner, R., and King, M. A.: Basal friction of Fleming Glacier, Antarctica, Part A: sensitivity of inversion to temperature and bedrock uncertainty, The Cryosphere, companion paper. companion paper.
Zhao, C., King, M. A., Watson, C. S., Barletta, V. R., Bordoni, A., Dell, M., and Whitehouse, P. L.: Rapid ice unloading in the Fleming Glacier region, southern

Antarctic Peninsula, and its effect on bedrock uplift rates, Earth and Planetary Science Letters, 473, 164-176, 2017.

---

## Author Comment (AC2) · 19 May 2018

We are grateful to Reviewer 2 for the positive and constructive suggestions to improve our paper. We have addressed the comments below. The line numbers in the responses are based on the revised manuscript without change track.

Please note that Mathieu Morlighem created the ice thickness data for the Fleming Glacier system using the mass conservation method, which is very important for most experiments done in this study. We do value his contribution to this paper, so we add him as the co-author in the revised text.

In the revised companion paper (Zhao et al., companion paper), we implemented a new sensitivity test to the enhancement factor (E). It reveals that the optimal value of E = 1.0 should be chosen as the enhancement factor in the CONTROL experiment. Accordingly, we re-ran all the simulations in this study with E = 1.0, and the high basal shear stress band near the ice front in 2008 has decreased into high basal shear spots, which are suspected of being artefacts of the inversion process and are discussed below. We modified the text and figures accordingly. All other result and interpretations are not qualitatively changed from the original manuscript.

General comments

This paper, using diagnostic inverse modeling of basal conditions, discusses the possible causes of the retreat of Fleming glacier observed between 2008 to 2015. In particular, the potential acceleration induced by the production of water by frictional heating at the base of the glacier is discussed. This paper is well written, even if some sentences are too long and some figures can be improved. I have made below some suggestions that I believe could improve the manuscript.

Specific comments

line 62: nearly twice or more than twice?

"More than twice" is more suitable here. Modified.

line 95: I don't really see where in Gladstone et al. (2017) inverse methods are used?

The reference is deleted here.

line 123: define what is bed_zc

bed_zc, has been defined using Eq. (1). To clarify it better, we modified the sentence into "The bedrock data, bed_zc (Fig. 2b), …" (Line 136)

line 127: S2008 is not the "surface DEM in 2008" but the "surface elevation in 2008".

We modified it into "where $S_{2008}$ is the surface elevation in 2008 combined from two DEM products as discussed in Zhao et al. (companion paper),… " (Line 140-141).

line 134: (on the same line) The "2008 velocity" should be "The 2008 velocity

dataset"

Modified.

line 155: the assumption that all the ice is grounded is for the inverse method? May be you can specify already here that floating ice will be deduced as the place where basal stress is lower than a threshold? It is not clear all along the manuscript if there is still a floating part or not on Fleming glacier and it would help if it could be mentioned more clearly in the introduction.

Yes, the assumption that all the ice is grounded is for the inverse method. The floating ice will be deduced where basal shear stress is lower than a threshold. To clarify this, we added a sentence "This assumption might be incorrect for the main branch of the FG, and we evaluate it based on the deduced floating area where the inferred basal shear stress is lower than a threshold, which is discussed in Sect. 4.1." (Line 172-175).

In the introduction, we declared that the ice front position in Apr 2008 (dark blue line in Figs. 1b and 1c, Wendt et al. (2010)) has almost coincided with the 1996 grounding line position (Line 62). For this study, we assume that all the ice is grounded and the ice front position is same as the 1996 ice front position, which is added in Line 171-172.

line 175: it should be mentioned that Eq. (4) is valid under the assumption of $N = 0$

Thanks for the suggestion. We added one sentence after this equation (Line 208-210). "Here we assume that the water pressure in the subglacial hydrologic system is given by the ice overburden pressure, which is equivalent to assuming that the effective pressure at the bed, $N$, is zero (Shreve, 1972)"

line 186: here it should be mentioned that Eq. (6) is derived under the assumption of a perfect connectivity of the basal hydrology system with the ocean

Thanks for the suggestion. We did say that we used a simpler hydrostatic balance. In order not to get tangled up with the interior hydraulic modeling, we add a sentence to qualify this "This expression for $Z_*$ assumes a perfect connectivity of the basal hydrology system with the ocean. This is appropriate for the present study where we are exploring the degree of grounding of the fast flowing regions of the FG over the downstream basin." (Line 217-220).

line 192: C is not a vector (not in bold)

Modified.

line 380: The increase of the amount of melt water should be quantified by integrating the frictional heating over the bedrock. But it should be also discussed that more melt doesn't necessarily induce an acceleration of the glacier as the basal hydrology system is evolving dynamically to adjust this surplus of water. The link of basal sliding with basal water should be clarified, and specifically is should be mentioned that the important variable is not the amount of water but its pressure. And this later quantity is not evaluated in the present work.

The amount of melt water has been quantified based on the Eq. S1 in the Sect. S2 and shown in Fig. S4 in the supplementary material. We present the distribution of the basal melt water along with a 2015-2008 difference plot rather than presenting the integrated total. This approach demonstrates the patterns and regions of important differences, which would not be apparent in an integrated quantity. Also, the

integrated basal melt would be sensitive to the region of integration. We mentioned this in Line 468-471.

We have clarified the positive feedback mechanism in Sect. 4.2 (Line 295-301). "Since the reduction of effective pressure is the key process to enhance sliding, this positive feedback is dependent on a positive feedback of melt water generation to water pressure. This dependence can break down when there is sufficient basal water to generate efficient drainage channels (Schoof, 2010). However, such efficient channelization in the subglacial hydrologic system is typically associated with seasonal surface meltwater pulses reaching the bed (Dunse et al., 2012), a process that is not expected to occur for Fleming Glacier (Rignot et al., 2005)."

For the subglacial water pressure, it is not possible to evaluate this quantity without a hydrology model, which is beyond the scope of this study.

line 430: Can the buttressing exerted by the pining band in 2008 be quantified in a more rigorous way? A complementary experience would be to remove this band of high friction (by setting no friction there) and to see how the velocity field is modified upstream. This would directly quantify the increase of velocity induced by an instantaneous loss of the pining band. The difference between this velocity field and the 2015 one would indicate places where a decrease of basal shear stress is necessary to explain the 2015 velocity field.

We integrated the basal shear stress (~3.42e11 N) for the frontal sticky spots in 2008 (where the Taob>0.01 MPa shown in Fig. S3). We have clarified this in Line 232.

We have tried some sensitivity tests to different ice front positions and ice front ocean-pressure boundary conditions in the companion paper (Zhao et al., companion paper). Those experiments have a similar effect to modifying basal shear stress near the ice front. The results show that those changes didn't impact on the velocity very far upstream. So this unpinning on its own is unlikely to have caused the speed up, but it could be a trigger for basal feedbacks to kick in.

line 528: Schaëfer is not spelled correctly

Modified.

Caption Fig. 1: inset (c) should be located in (b) and in (c) the front position in

2008 and 2016 should be added to visualise a potential ice-shelf?

Modified and added.

Fig. 3: the grounding line in 2014 seems to have a different form than the one of Friedl et al. (2017) in their Fig. 6?

Fig. 3 is generated with Paraview. To add the grounding line of 2014 in Paraview, we have to generate the mesh with the grounding line of 2014. A typical element size in this region is ~200-300 m. The only difference between the grounding line in Fig. 3 and the original shapefile is mapping it to nodes on the Elmer mesh, therefore the differences are always less than one element size. The mesh size and the refinement affected the location of grounding line. So the difference is never more than 300 m (an element's width), and it would not affect the analysis in this paper.

References

Dunse, T., Schuler, T. V., Hagen, J. O., and Reijmer, C. H.: Seasonal speed-up of two outlet glaciers of Austfonna, Svalbard, inferred from continuous GPS measurements, The Cryosphere, 6, 453-466, 2012.

Rignot, E., Casassa, G., Gogineni, S., Kanagaratnam, P., Krabill, W., Pritchard, H., Rivera, A., Thomas, R., Turner, J., and Vaughan, D.: Recent ice loss from the Fleming and other glaciers, Wordie Bay, West Antarctic Peninsula, Geophysical Research Letters, 32, 2005.

Schoof, C.: Ice-sheet acceleration driven by melt supply variability, Nature, 468, 803-806, 2010.

Shreve, R.: Movement of water in glaciers, Journal of Glaciology, 11, 205-214, 1972.

Wendt, J., Rivera, A., Wendt, A., Bown, F., Zamora, R., Casassa, G., and Bravo, C.: Recent ice-surface-elevation changes of Fleming Glacier in response to the removal of the Wordie Ice Shelf, Antarctic Peninsula, Annals of Glaciology, 51, 97-102, 2010.

Zhao, C., Gladstone, R., Zwinger, T., Warner, R., and King, M. A.: Basal friction of Fleming Glacier, Antarctica, Part A: sensitivity of inversion to temperature and bedrock uncertainty, The Cryosphere, companion paper. companion paper.

---

## Author Comment (AC3) · 19 May 2018

We are grateful to Reviewer 3 for the positive and constructive suggestions to improve our paper. We have addressed the comments below. The line numbers in the responses are based on the revised manuscript without change track.

Please note that Mathieu Morlighem created the ice thickness data for the Fleming Glacier system using the mass conservation method, which is very important for most experiments done in this study. We do value his contribution to this paper, so we add him as the co-author in the revised text.

In the revised companion paper (Zhao et al., companion paper), we implemented a new sensitivity test to the enhancement factor (E). It reveals that the optimal value of E = 1.0 should be chosen as the enhancement factor in the CONTROL experiment. Accordingly, we re-ran all the simulations in this study with E = 1.0, and the high basal shear stress band near the ice front in 2008 has decreased into high basal shear spots, which are suspected of being artefacts of the inversion process and are discussed below. We modified the text and figures accordingly. All other result and interpretations are not qualitatively changed from the original manuscript.

GENERAL COMMENTS

Main question in the abstract: Is the observed acceleration of the flow and thinning of the glacier due to increased ocean warming and/or marine ice sheet instability?

Method: Infer basal shear stress from observations and calculate a steady state temperature field using a Stokes ice sheet model for 2008 and 2015.

Results: Reduction in magnitude and increase in area of low basal shear stress near the 1996 grounding line and reduction in height above flotation between 2008 and 2015 suggest the grounding line has retreated for Fleming Glacier, southern branch of Fleming Glacier and Prospect Glacier.

A band of higher basal shear stress parallel to the 1996 grounding line at 2008 suggests that Fleming Glacier was still grounded at that time. Subglacial water may be generated from high basal frictional heating upstream of Fleming Glacier. Frictional heating has increased between 2008 and 2015 over a rise between two deep bedrock basins.

As mentioned above, in the revised companion paper, we implemented a new sensitivity test to enhancement factor (E). It reveals that the optimal value of 1.0 should be chosen as the enhancement factor in the CONTROL experiment. So we redid all the simulations in this study with E=1.0, and the high basal shear stress band near the ice front changed into high basal shear spots in 2008, which are suspected to be artefacts. We did not rule out, however, the possibility that the ice front was still grounded on some pinning points. We discuss this point in the first sentence of the Discussion section (Line 359-361).

Comments: I don't think the main question can be answered from instantaneous time slices of the ice flow. The authors need to do forward experiments with various ocean forcing such as different basal melt rates or vertical melting at the calving front. Alternatively, the authors need to pose a different question. The band of high basal shear stress may not be physical realistic. The model error reported in their companion paper is relatively high in this area.

Clearly forward modelling of the Fleming system to study the recent ungrounding transition is a natural next step. That would, as the reviewer acknowledges, require an extensive exploration of forcing influences. In the present work we have clearly shown the differences in basal shear stress distributions for the Fleming system between 2008 and 2015, reflecting different surface elevations and the recent acceleration in ice flow. This has provided insights into the recent ungrounding – and suggested possible feedback processes that may have contributed to the recent changes. We consider this scope has provided sufficient worthwhile material for the present paper. Experiments with future coupled ice sheet-ocean models would also be valuable. We have mentioned this in the Conclusion section (Line 535-537).

In the modified companion paper (Zhao et al., companion paper), the misfit between the simulated and observed surface velocity at the ice front of the FG is very small. The difference between the relaxed and observed surface is < 15 m after three cycles in the CONTROL experiment. It means the modified model with the enhancement factor of 1.0 models the ice front well.

Interesting idea: The authors propose that basal water generated from high basal frictional heating upstream draining towards the front, triggered grounding line retreat of Fleming Glacier. This mechanism is an alternative to the usual ocean forcing explanation. Mass loss could significantly increase, due to marine instability, if the grounding line retreated over a bedrock rise into the second deeper basin. The highest frictional basal heating in 2015 is located over the rise, which may be a potential trigger for the grounding line retreat.

Manuscript in general: The font is too small and the text is not double spaced, which made reviewing the paper tricky. Picking out the references was particularly difficult give the font size and text spacing. Some of figures are too small.

Apologies if the manuscript was not in the format the Reviewer expected. We are happy to comply with whatever formatting requests are made by the Copernicus staff in this regard.

SPECIFIC COMMENTS

Ocean forcing: It seems reasonable to suggest that increased melting at the vertical face of the front of FGL due to incursions of CDW may have affected the pressure boundary condition at the front sufficiently to remove the high band of basal shear stress. However, I don't think your results shed any new light on what has been suggested in the other references you use about ocean basal melting. Forward time-dependent modelling experiments are needed to test these theories and here's an example for Larsen B of how you can extend the work you have done for this paper. Vieli et al 2007 Causes of pre-collapse changes of the Larsen B ice shelf: Numerical modelling and assimilation of satellite observations. Earth and Planetary Science Letters. https://doi.org/10.1016/j.epsl.2007.04.050

As we mentioned above, we agree that transient experiments will be valuable, but are beyond the scope of the current study. We aim to carry out both transient ice dynamic

simulations and coupled ice-ocean simulations, and hope we will be able to bring such studies to fruition over the coming years.

Grounding line retreat: The results for 2015 of low basal shear stress and low height above buoyancy confirm the findings of Friedl et al 2017 that Fleming Glacier's grounding line has retreated. The results for PGL are different to FGL: Driving stress appears to be much higher for PGL in 2015.

The revised ratio of driving stress $\tau_{d2015}$ to $\tau_{d2008}$ (Fig. 3f) shows that the driving stress of PG in 2015 was much lower (not higher) than 2008. We have clarified that the cases for the southern FG and PG are different from the main branch of FG. We did not account in the model for the remaining ice shelf for those two glaciers because we do not have the ice thickness data for the ice shelf. We modified our analysis on those two glaciers (Line 275-284). We think the northern section of the southern FG has been ~2 km behind the 1996 grounding line position based on the ice front position shown in Fig. 1c. However, it is hard to decide whether the southern section of the southern FG or the PGL have also retreated from 2008 (Fig. 3a) to 2015 (Fig. 3b), since we did not account for the normal stress of the remaining small ice shelf at the front of the southern FG (Fig. 1c) in the inverse modelling. Note that the hypsometric model used generate the DEM in 2015 is based on the observed elevation change rates (Zhao et al., 2017). However, the observations are mainly focused in the FG region (Fig. 2a), so the DEM2015 of PG could be an artefact. That might explain why the driving stress was lower in 2015 (Fig. 3f).

Temperature homologous near the 1996 grounding line (for PG) appears much lower in 2015 suggesting that the glacier may have become frozen to the bed?

Note that we have replaced the term "temperature homologous" with "temperature relative to pressure melting point" in the entire text.

The temperature near the ice front/grounding line of PG is indeed colder in the 2015 steady-state calculation. The main difference in the modeled temperature between 2008 and 2015 is due to a reduction in friction heat. This is in turn due to reduced basal shear stress, which occurs in the inversion as a result of the reduced driving stress compared to 2008. This may be due to the lack of observational hypsometric data – the imposed surface lowering (which causes the driving stress reduction) in 2015 is based mainly on data from FG.

A contributing factor could be the steady state temperature assumption, which is almost certainly worse for 2015 than it is for 2008, because the recent acceleration means that the glacier is further from steady state in 2015 than in 2008.

Also, the current modelling approach does not represent the capacity of the subglacial hydrologic system to redistribute heat at the bed. In reality the flow of basal melt water from upstream to downstream will bring more latent heat to the base of the ice sheet near the grounding line.

Is the band of high basal stress at the front of FGL physically realistic? The authors attempt to address this question in the paragraph beginning on line 209. Part A shows that the misfit between the modelled and observed speed is high, where the modeled speed is too fast, and the surface slope is also higher here than over the region of low shear stress. The driving stress is not obviously high given the relatively high surface slope. What concerns me is your model appears unable to model the front.

As mentioned above, the revised companion paper of this study (Zhao et al., companion paper) shows high basal stress spots rather than a band (as previously) at the front of FGL. This may, as the reviewer suggests, be an artefact, owing to various uncertainties. We also do not rule out the possibility that the ice front was still grounded on some pinning points. We clarified this in Line 359-361.

In the revised companion paper (Zhao et al., companion paper), the misfit between the simulated and observed surface velocity at the ice front of the FG has been very small. The difference between the relaxed and observed surface is < 15 m after three cycles in the CONTROL experiment. It means the modified model with the enhancement factor of 1.0 models the ice front in 2008 better now than in the version of the companion paper to which the reviewer refers. It is also worth noting that our 2015 simulations have not had any difficulties modelling the ice front. This suggests that the problem is in the boundary conditions rather than the model itself, which was the motivation for the ice front position and pressure sensitivity experiments in the companion paper. These experiments indicated that the inversion is only sensitive to such ice front uncertainties within a short distance of the front.

What about rheology of the ice near the front? Perhaps the standard A is not appropriate here. Part A shows a large vertical shear at the front where the basal speed is much smaller than the surface. Is the ice stiffer at the front? Vieli et al 2006 Numerical modelling and data assimilation of the Larsen B ice shelf, Antarctic Peninsula, Phil. Trans. R. Soc. A, 364, 1815–1839, doi:10.1098/rsta.2006.1800 solved the inversion problem for effective viscosity. Modelling a front is difficult!

Based on the sensitivity test to various values of enhancement factors (0.5, 1.0, 2.0, 4.0) in the revised companion paper, we found that the value of 1.0 is the optimal value for the overall Fleming system. Various studies of anisotropic ice properties and enhancement factors (e.g. Graham et al. (2018); Ma et al. (2010)) suggest that ice near the ice front could well be stiffer than ice deforming under simple shear near the bedrock in the interior of the ice sheet, however, we have only a uniform enhancement factor E as a control parameter in the present study.

About solving the inversion problem for effective viscosity: it is simple to invert for ice rheology in an ice shelf model, as suggested. Here for the grounded glacier - certainly largely grounded in 2008 - the velocity mismatch can be addressed by adjusting the ice stiffness and the basal drag. Simultaneous inversions for stiffness and basal friction coefficient are possible but beyond the model tools we have available.

What about the direction of the flow? Is there a difference is the modelled flow direction and the observed direction? Is there a change in flow direction between 2008 and 2015 as the ice moves over the sticky band and becomes ungrounded. Also, could ice melange at the front FGL affect the boundary condition?

The inversion scheme we used (following Gagliardini et al. (2013)) only compares the mismatch in modelled and observed speeds, not directions. To a simple visual inspection the velocity directions in 2008 and 2015 are very similar if not identical.  A direct overlay of streamlines may allow minor deviations to be identified, but we have not identified an urgent need to such analysis to be carried out.
About the ice mélange at the ice front, we explored the effect of an extra normal force at the ice front (to simulate the potential effect of ice mélange) in the ice front boundary condition experiments of the revised companion paper (Zhao et al.,

companion paper). We calculated that ice mélange back force (~1.1e7 N m$^{-1}$) used to prevent the rotation of iceberg at the calving front (Krug et al., 2015) could account for the equivalent of up to ~2.3 m sea level in terms of ice front boundary condition, which was included in the experiments with different sea levels.

Basal frictional heating and subglacial water: Could the region of low basal shear stress near the front simply be due to subglacial water from upstream pooling in the bedrock basin, e.g. FGL in 2008? Could the region be partially grounded?

The reviewer may be right. Figs. 4d and 4e show that there is a plateau in the hydraulic potential in the downstream basin. In 2015, we think the downstream basin is mainly ungrounded, based on the inferred basal shear stress (Fig. 3b) and the height above buoyancy (Fig. 5b). There could therefore be some pooling of water in 2008, and it could be partially grounded, but a big cavity is not possible given the geometry. Our inferred basal shear stress (Fig. 3a) and the height above buoyancy (Fig. 5a) show that the downstream basin of the FG in 2008 should still remain grounded.

In the discussion section, we have clarified the issue of basal hydrology, and the potential of water to pool at certain locations (Line 471-476). "The plateaus in hydraulic potential in both downstream and upstream basins of the FG suggest the possibility that basal water may accumulate in those regions, or at least show a low throughput. The downstream plateau appears to be fed by a large frictional heat source over the ridge between the downstream and upstream basins in addition to flow from further inland, while the upstream plateau appears to be fed by an extensive upstream region of basal melting. "

The temperature homologous is high, which prevents the water from refreezing. I think the role of subglacial water could be explained more in the literature review. Paragraph beginning 85:

We modified this sentence (Line 98-102) into "A positive feedback between basal sliding and basal water pressure (through friction heating) upstream of the grounding line could be another possible factor in the glacier acceleration and grounding line retreat (Bartholomaus et al., 2008; Iken and Bindschadler, 1986; Schoof, 2010). The possibility of such a feedback, is not ruled out by Friedl et al. (2018), and is discussed further in Sect. 4.2 and Sect. 5."

You don't explain what the feedback mechanism is. As I understand it, Schoof (2010) talks about the importance of variability of basal water on flow dynamics, with flow accelerating due to a short-lived increase in basal water, but then the flow slows if the basal water stays high. Is that happening here? You have high basal frictional melting in 2015, which you say is speeding up the flow, but figure 3 in Friedl et al 2017 shows that the ice speed of FGL decreased between 2011-2015.

We have added a description about the positive feedback in Sect. 4.2 (Line 288-301). As we have clarified, "Since the reduction of effective pressure is the key process to enhance sliding, this positive feedback is dependent on a positive feedback of melt water generation to water pressure. This dependence can break down when there is sufficient basal water to generate efficient drainage channels (Schoof, 2010). However, such efficient channelization in the subglacial hydrologic system is typically associated with seasonal surface meltwater pulses reaching the bed (Dunse et al., 2012), a process that is not expected to occur for Fleming Glacier (Rignot et al., 2005)."

In the published Fig. 2 and Fig. 3 in Friedl et al. (2018), it shows that the ice speed of

FGL remains stable with a very small median velocity increase (0.06 m d$^{-1}$) from 2011 to 2016, not decrease. Besides, the speed up in 2015 is relative to the surface velocity 2008. We do not dispute the observed acceleration phrase occurred in Mar 2010-early 2011 found by Friedl et al. (2018).

Figure 1 shows that the ice front for PGL and sFGL has calved between 2008 and 2016, with some advance in the southern part of the bay. Could calving event(s) explain the speed up and lowering of the surface of the streams? Also, from figure 1, PGL has an ice shelf, are you applying the normal stress, hydrostatic pressure boundary condition at the 1996 grounding line or at the real front?

Yes, we agree with the reviewer. The calving events may explain the speed up of the PGL and sFGL. The surface lowering for those two regions have not been confirmed by Zhao et al. (2017) owing to the lack of elevation observations. But inversions will not give a clear answer to this - transient experiments would be needed. A good experiment to do would be to carry out an inversion with an advanced ice shelf, then two transient experiments: one simply carrying on from the inversion, and one in which the shelf is removed. The difference between these transient experiments would be informative as to the impact of calving on flow speeds and surface lowering. Such experiments would be interesting, and we hope to have a chance to carry out such experiments, but that they are beyond the scope of the current study.

As we responded above, we did not account for the remaining ice shelf for those two glaciers because we don't have the ice thickness data for the ice shelf. We modified our analysis on those two glaciers (Line 275-284).

Bedrock plots: The way the bedrock is plotted is a bit inconsistent and unclear. Figure 1 c clearly shows where bedrock is above or below sea level with a white colour band around 0, but the bed elevation colour in figure 4 c and d looks like most of the bedrock is either below or at sea level and figure 2 b is too small. It would be useful to see where the retrograde and prograde slope are.

We modified Fig. 2b to have the same color scale as Fig. 1c. For Figs. 4d, 4e (original Figs. 4c, 4d), we think it is better to use a different color scale to show the retrograde and prograde slopes. Here we plotted the bed elevation with meters above sea level. The regions of retrograde slope are now easily identifiable by eye in Figs. 4d, 4e, but plotting these regions is not trivial to automate.

Figures 3,4,5: useful to have a third column of figures showing the difference between the first two columns.

Thanks for the suggestion. For Fig. 3, we computed the ratio of $\tau_{b2015}$ over $\tau_{b2008}$ (Fig. 3e), and also the ratio of $\tau_{d2015}$ over $\tau_{d2008}$ (Fig. 3f) to represent the difference. For Fig. 4 and Fig. 5, we plotted the difference between 2015 and 2008 (2015 minus 2008).

Line 158: The Linear sliding law is fine for the inverse problem because $\tau$ remains unchanged, if a higher power of u is used, only the coefficient C would change. However, for a forward run a linear law may be inappropriate.

We agree. There is, however, no transient simulation involved in this study except in the brief surface relaxation step. For the steady-state temperature simulation, the Stokes solver is turned off and the velocity field is fixed. It means that the basal shear stress is fixed for the temperature simulation. So we don't think it is inappropriate to run our simulations with a linear sliding law. For transient simulations we intend to

convert *C* to a distribution that, for whatever sliding law is used, gives the same initial basal shear stress distribution as we obtain from the inversion.

Please define basal frictional heating in your method section.

We have added the equation as Eq. 4.

Figure 2 is too small and/or too detailed. A difference plot of surface elevation may be more informative.

Modified.

Figure 3: The patterns in c and d seem to be influenced by the computational mesh. Have you investigated mesh resolution by halving or doubling element sizes?

The sensitivity tests to horizontal (1 km, 500 m, 250 m, 125 m) and vertical (10 layers and 20 layers) mesh resolution have been carried out in our companion paper (Zhao et al., companion paper). The resolution of 250 m is fine enough for inverse modeling in this study. Also note that the features in Figs. 3c and 3d (now 3d and 3e after revision) are much coarser than the element size – these features are resolved.

The ratio of basal shear stress to driving stress: I'm not sure what figure 3 c and d are showing or why a low value means the ice may be close to flotation.

A balance between tau_b and tau_d is indicative of a heavily grounded regime – longitudinal stresses are low and the dominant force balance is between the gravitational driving stress and the basal resistance. This assumption is similar to the assumption behind the derivation of the "shallow ice approximation", which has been used extensively for long time scale grounded ice sheet simulations. The opposite assumption, the "shallow shelf approximation" (SSA), is a balance between membrane stresses and driving stresses, with basal shear stress being vanishingly small. This is the typical stress balance in a floating ice shelf. So a low value of this stress ratio means we're closer to the SSA regime, i.e. an ice shelf regime.

Figure 4 c and d: Would showing the potential gradient be more informative? I can't see labels on the contours of hydraulic potential.

We agree that the direction of flow is important, but a vector plot of the potential gradient would be rather complicated. The direction of flow, perpendicular to the contours of hydraulic potential, should be clear from the figure, with very few local minima. The magnitude of the gradient can be clearly seen in the proximity of contours of hydraulic potential. Where the contours are close together is where the gradient is steep. Labelling the contours increases the complexity of the plot, and in fact the main purpose of the whole calculation is to demonstrate the pattern of basal water flow rather than to estimate specific values. We have actually tried several options for how to present this Figure, including a scalar field plot of hydraulic potential gradient, but we are still confident that our current combination of bedrock and hydraulic potential contours gives the clearest picture of the pattern of basal water flow.

Figure 5: Height above buoyancy appears to be negative south of the main stream of FGL (and south of PGL). Is the bedrock above sea level there?

No. Note that the height above buoyancy is never negative where the bedrock is above sea level (Eq. 7). Negative values simply indicate that the estimated thickness of ice present should be afloat, given the bedrock is so far below sea level. As we discuss in the text, uncertainties in ice thickness and bedrock data affect the

calculation of the height above buoyancy. The bedrock of the southern branch of the FG is below sea level while a little part of the south PG is below sea level. To clarify the question of bedrock values, we modified the Fig. 1c to show the bedrock below sea level only.

Discussion section: Is a maximum melt rate of 1 m/a enough to generate a plume of high enough velocity to entrain incursions of CDW to enhance basal melting beneath the floating ice? You can calculate the flux of subglacial water for each year by Taobub/Lii x area that feeds the grounding line based on the hydraulic potential (or its gradient).

To address this question, we need a 3D ocean model. The buoyant plume is a function of many things, of which subglacial outflow is only one. We'd need to know a lot about the CDW pathways in the area, whether CDW is coming into contact with the grounding line, volume fluxes, heat fluxes, the regional oceanography. So yes, calculating total subglacial outflow is relatively straightforward but simulating the local ocean circulation and plume behavior is way beyond the scope of the current study, and the subglacial outflow is a fairly meaningless number without this oceanographic context.

Line 395: Could you explain the positive feedbacks. Estimating the time scale for the ice to unground from the rise between basins leaving the ice stream vulnerable of marine instability in the upstream basin is good, but I'm not sure you can say height above buoyancy is a measure of potential mass loss.

We have added a description of the positive feedbacks in Sect. 4.2 (Line 288-301).

We did not say the height above buoyancy could indicate potential mass loss. It indicates potential vulnerability. If $Z^*$ is close to zero, then the system is close to ungrounding, and would only require a small perturbation to unground. The height above buoyancy would be relevant to sea level rise in a grounding line retreat situation. The accelerated ice flux across the grounding line is probably a separate issue.

TECHNICAL CORRECTIONS

Line 46: abbreviation 'GL' is not defined.

We don't use the abbreviation "GL" for the "grounding line", so we modified "GL" into "grounding line".

Line 88: Not sure the sentence is helpful. Might be better to delete it.

Deleted.

Section 2.2: Is Hmc part of a dataset from Morlighem or have you combined two dataset yourself?

Morlighem has been added as the co-author of both the companion paper and this paper. He generated Hmc for the companion paper (Zhao et al., companion paper). Hmc includes three regions: for fast flowing region, he computed the ice thickness data for fast-flowing regions using the Ice Sheet System Model's mass conservation method (Morlighem et al., 2011; Morlighem et al., 2013), based on ice thickness measurements from the Center for Remote Sensing of Ice Sheets (CReSIS), using ice surface velocities in 2008 from Rignot et al. (2011b), surface accumulation from RACMO 2.3 (van Wessem et al., 2016) and 2002-2008 ice thinning rates from Zhao et al. (2017); for slow flowing region, he adopted data from bedmap2; for the

transition region, he smoothed the data. It has been clearly described in the revised companion paper.

Lines 132, 151: Part 1 or Part A

Thanks for pointing this out. It should be "Part A".

Line 145: Is the basal frictional heating calculated from output from the inverse problem and used as an input into the heat equation?

Yes. The basal frictional heating shown in Figs. 4a and 4b, is calculated directly from the output of the inversion process – as the product of inferred basal shear stress and basal velocity – see Eq. 4, which we added at the request of this reviewer (see above). The basal frictional heating is an integral part of the steady state temperature simulation. That calculation does use the velocities and friction coefficients from the inversion, so the frictional heating from the inversion is indeed included.

184: I don't think N needs a numbered equation because it isn't used.

We would like to make this change if the Copernicus proofreaders request it.

Line 222: northern and eastern. It might be helpful to add an arrow indicating North on one of the figures.

Added.

Lines 367, 420: Friedl et al 2017 gave an estimated grounding line for 2014.

Thanks for pointing this out. We have modified the relevant text to grounding line in 2014.

Figure 1: sFGL is not marked on the figure.

Added.

Figure 3: It is difficult to work out where the plotted regions exist in relation to figures 1 and 2. Orientation is given in figure 5 but would be more useful on figure 3.

Thanks for the reviewer's suggestions. We have added an inset map in Fig. 3a to show the plotted region. We also added the north direction in Fig. 3a.

Figure 3: Cannot see cyan contour on printed paper.

We modified the color of all the velocity contours into white color.

Figure 4: I can't distinguish between red and magenta contours.

We changed both colors in Fig. 3a, 3b, 3d, 3e.

Line 529: Case is wrong for Schafer.

Modified.

References
Bartholomaus, T. C., Anderson, R. S., and Anderson, S. P.: Response of glacier basal motion to transient water storage, Nature Geosci, 1, 33-37, 2008.
Dunse, T., Schuler, T. V., Hagen, J. O., and Reijmer, C. H.: Seasonal speed-up of two outlet glaciers of Austfonna, Svalbard, inferred from continuous GPS measurements, The Cryosphere, 6, 453-466, 2012.
Friedl, P., Seehaus, T. C., Wendt, A., Braun, M. H., and Höppner, K.: Recent dynamic changes on Fleming Glacier after the disintegration of Wordie Ice Shelf, Antarctic Peninsula, The Cryosphere, 12, 1-19, 2018.

Gagliardini, O., Zwinger, T., Gillet-Chaulet, F., Durand, G., Favier, L., de Fleurian, B., Greve, R., Malinen, M., Martín, C., Råback, P., Ruokolainen, J., Sacchettini, M., Schäfer, M., Seddik, H., and Thies, J.: Capabilities and performance of Elmer/Ice, a new-generation ice sheet model, Geosci. Model Dev., 6, 1299-1318, 2013.

Graham, F. S., Morlighem, M., Warner, R. C., and Treverrow, A.: Implementing an empirical scalar constitutive relation for ice with flow-induced polycrystalline anisotropy in large-scale ice sheet models, The Cryosphere, 12, 1047-1067, 2018.

Iken, A. and Bindschadler, R. A.: Combined measurements of Subglacial Water Pressure and Surface Velocity of Findelengletscher, Switzerland: Conclusions about Drainage System and Sliding Mechanism, Journal of Glaciology, 32, 101-119, 1986.

Krug, J., Durand, G., Gagliardini, O., and Weiss, J.: Modelling the impact of submarine frontal melting and ice mélange on glacier dynamics, The Cryosphere, 9, 989-1003, 2015.

Ma, Y., Gagliardini, O., Ritz, C., Gillet-Chaulet, F., Durand, G., and Montagnat, M.: Enhancement factors for grounded ice and ice shelves inferred from an anisotropic ice-flow model, Journal of Glaciology, 56, 805-812, 2010.

Morlighem, M., Rignot, E., Seroussi, H., Larour, E., Ben Dhia, H., and Aubry, D.: A mass conservation approach for mapping glacier ice thickness, Geophysical Research Letters, 38, n/a-n/a, 2011.

Morlighem, M., Seroussi, H., Larour, E., and Rignot, E.: Inversion of basal friction in Antarctica using exact and incomplete adjoints of a higher-order model, Journal of Geophysical Research: Earth Surface, 118, 1746-1753, 2013.

Rignot, E., Casassa, G., Gogineni, S., Kanagaratnam, P., Krabill, W., Pritchard, H., Rivera, A., Thomas, R., Turner, J., and Vaughan, D.: Recent ice loss from the Fleming and other glaciers, Wordie Bay, West Antarctic Peninsula, Geophysical Research Letters, 32, 2005.

Rignot, E., Mouginot, J., and Scheuchl, B.: Ice Flow of the Antarctic Ice Sheet, Science, 333, 1427-1430, 2011b.

Schoof, C.: Ice-sheet acceleration driven by melt supply variability, Nature, 468, 803-806, 2010.

van Wessem, J. M., Ligtenberg, S. R. M., Reijmer, C. H., van de Berg, W. J., van den Broeke, M. R., Barrand, N. E., Thomas, E. R., Turner, J., Wuite, J., Scambos, T. A., and van Meijgaard, E.: The modelled surface mass balance of the Antarctic Peninsula at 5.5 km horizontal resolution, The Cryosphere, 10, 271-285, 2016.

Zhao, C., Gladstone, R., Zwinger, T., Warner, R., and King, M. A.: Basal friction of Fleming Glacier, Antarctica, Part A: sensitivity of inversion to temperature and bedrock uncertainty, The Cryosphere, companion paper. companion paper.

Zhao, C., King, M. A., Watson, C. S., Barletta, V. R., Bordoni, A., Dell, M., and Whitehouse, P. L.: Rapid ice unloading in the Fleming Glacier region, southern Antarctic Peninsula, and its effect on bedrock uplift rates, Earth and Planetary Science Letters, 473, 164-176, 2017.

---

## Author Response (AR2)

Editor's comments

We thank the Editor Ben Smith for the positive and constructive suggestions to improve our paper. We have addressed the comments below.

Comments to the Author:

Editor's note on tc-2017-242

Most of the referees' comments are addressed in the response. I think that some clarification is in order in at least one case, and that one of the figures need to be a little easier to look at. If these changes can be made, I don't think the manuscript needs a second round of reviews.

Revisions requested:

The referees had trouble with the term 'dominant,' and I also had trouble understanding what it meant in the manuscript. The authors, I think, try to establish that their feedback-driven change in basal shear stress is possibly sufficient to explain the changes in glacier speed, which would imply that ongoing enhanced melt is not necessary to explain the changes in the glacier. But this does not establish one process or the other as dominant. I think a little bit of clarification early in the paper would help.

Thanks for the Editor's suggestion. Basal melting driven by ocean warming or the continued ice dynamic thinning combined with a bedrock unpinning could be possible triggers for the recent glacier acceleration and grounding line retreat. For each case, the proposed positive subglacial hydrological feedback may have played an important role in the ongoing changes. To clarify this, we modified the sentence in the conclusion section from "In either case, feedbacks in the subglacial hydrologic system may provide the dominant mechanism for rapid increases in basal sliding and ongoing ungrounding." into "In either case, feedbacks in the subglacial hydrologic system may be a significant factor in reducing basal shear stress, leading to rapid increases in basal sliding and ongoing ungrounding (Line 530-532)."

To make it clear early in the paper, we modified the sentence in Sect. 1 "An alternative hypothesis is that the recent changes arise from feedbacks in the dynamics of the evolving glacier, possibly involving the subglacial hydrology" into "An alternative hypothesis is that the recent changes are reinforced by feedbacks in the dynamics of the evolving glacier, possibly involving the subglacial hydrology (Line 84-86)". We also added a sentence there about another possible triggering mechanism at Line 83-84, "The recent acceleration could also be triggered by the continued dynamic thinning passing some threshold." This is discussed in Sect. 5 and concluded in Sect. 6.

Various factors (separately or in combination) could be behind the glacier acceleration and grounding line retreat, as we stated in the last paragraph of Sect. 5. Here we modified "the dominant cause of the recent FG ungrounding" into "the cause of the recent FG ungrounding" (Line 513-514). We also modified "Further research is necessary to better understand the dominant mechanisms." into "Further research is necessary to better understand the interplay of a range of possible mechanisms" (Line 514-515).

Figure 4: The figure numbers are illegible. The letters should be in a large font, in black, on a white background. In the left and middle columns, indicate the year as well. This is done much better in figure 5.

Thanks for the suggestion. The subplot labels in Figure 4 and Figure 3 have been revised to match figure 5, and the year labels added where appropriate as suggested.

Reviewer 3 was correct that Figs 4d and 4e are very difficult to interpret. The lines are far too thin and the colors used to indicate the basal elevation obscure the contours badly. The differences in hydraulic potential between the 2008 and 2015 fields are not visible in 4d and 4e, amounting (4f) to only 1-2 contours' difference. Since the text only mentions 4f in passing, and because 4d and 4e are visibly identical, but hard to read, I recommend removing 4d-4f, and generating a single-panel figure showing only the hydraulic potential in 2015 (or 2008), with either a color table or contour labels to indicate the hydropotential, and describing the changes between the two years with words.

Thanks for the Editor's suggestion. We deleted the original Figs. 4d-4f and took the hydraulic potential in 2008 as a separate new figure (Fig. 5). Here we generated the color plot of hydraulic potential in 2008 as the background to give the magnitude information, and kept the black contours to indicate clearly the subglacial water flow directions, which are orthogonal to the contours of hydraulic potential (Fig. 5a). The closeness of the contours conveys a clear sense of where the gradient is steep. To emphasize the role of bedrock elevations below sea level and the bedrock basins in the main Fleming Glacier in relation to the plateaus in hydraulic potential as we mentioned in Sect. 5 (Line 476-487), we retained the figure with bedrock below the sea level as the background shading under the same set of contours of hydraulic potential (Fig. 5b). The relevant text was also modified.

We agree with the Editor about deleting the original Fig. 4f and added one sentence about the changes of the hydraulic potential between 2008 and 2015 in Sect. 4.2, "The hydraulic potential evolves between 2008 and 2015 due to the changes in surface elevation (Fig 2a) in Eq. 5, but this does not appreciably change the pattern of subglacial water flow." (Line 321-323).

[revised manuscript text omitted]
. 4e, 4f) forin both epochs (Figs. 4d, 4e), and the contours of hydraulic potential in 2008 ($\Phi$; Fig. 5). Friction heating due to sliding at the bed (Figs. 4a, 4b) provides a basal melt water source where ice is at pressure melting temperaturepoint, which is the case for the fast flow regions of the FGL (see the basal homologous temperature relative to the pressure melting point in Figs. 4e4d, 4f4e), and while the gradient of the hydraulic potential (Figs. 4e5, 4d) indicates likely water flow paths at the ice-bed interface. The hydraulic potential evolves between 2008 and 2015 due to the changes in surface elevation (Fig 2a) in Eq. 5, but this does not appreciably change the pattern of subglacial water flow. 
[revised manuscript text omitted]
 2015 (rightmiddle). The differences of (c) basal friction heating and (f) simulated basal temperature between 2008 and 2015 (2015 minus 2008). The white dotted line represents the deduced grounding line in 2014 from Friedl et al. (2018). The white solid lines represents the 2008 surface speed contours of 100 m yr$^{-1}$, 1000 m yr$^{-1}$, and 1500 m yr$^{-1}$.

[Figure]

Figure 5. (a) The hydraulic potential in 2008 and (b) the submarine bedrock elevation (metres above sea level). In both figures the dense contours represent the hydraulic potential with a spacing of 20 m (black solid lines). The white dotted line represents the deduced grounding line in 2014 from Friedl et al. (2018). The white solid lines represent the 2008 surface speed contours of 100 m yr$^{-1}$, 1000 m yr$^{-1}$, and 1500 m yr$^{-1}$.

The magenta and red solid lines show the boundaries of area with $\tau_b$ < 0.01 MPa and area with RBD < 0.1, respectively. A and B indicate the location of two over-deepened regions in the downstream basin.

[Figure]

Figure 56. The height above buoyancy $Z_*$ in (a) 2008 and (b) 2015 of the Fleming Glacier and Prospect Glacier. The background images are from (a) ASTER L1T data in Feb 2nd, 2009, and (b) Landsat-8 in Jan 13th 2016, respectively. The black lines represent velocity contours in 2008 (Rignot et al., 2011c) and 2015 . The dashed black and blue lines show the grounding line in 1996 (Rignot et al., 2011a) and 2014 (Friedl et al., 2018), respectively. The dashed magenta line shows the possible grounding line with $Z_* < 20$ m. Inset map shows the location in the research domain with blue points showing the available elevation data points used to extract the hypsometric model of elevation change from 2008 to 2015 (Zhao et al., 2017).

[Figure]

Figure A1. Basal shear stress, $\tau_B$, for (a) 2008, (b) 2015, and (c) a simulation using topography from 2008 and velocity from 2015. The white dotted line represents the grounding line in 2014 estimated by Friedl et al. (2018). The black, yellow and cyan solid lines represent the 2008 surface speed contours of 100 m yr$^{-1}$, 1000 m yr$^{-1}$, and 1500 m yr$^{-1}$, respectively.